# Favipiravir antiviral efficacy against SARS-CoV-2 in a hamster model

Jean-Sélim Driouich [1,5], Maxime Cochin[1,5], Guillaume Lingas[2], Grégory Moureau[1], Franck Touret [1], Paul-Rémi Petit[1], Géraldine Piorkowski[1], Karine Barthélémy[1], Caroline Laprie[3], Bruno Coutard[1], Jérémie Guedj [2], Xavier de Lamballerie [1], Caroline Solas[1,4] & Antoine Nougairède [1✉]

Despite no or limited pre-clinical evidence, repurposed drugs are massively evaluated in clinical trials to palliate the lack of antiviral molecules against SARS-CoV-2. Here we use a Syrian hamster model to assess the antiviral efficacy of favipiravir, understand its mechanism of action and determine its pharmacokinetics. When treatment is initiated before or simultaneously to infection, favipiravir has a strong dose effect, leading to reduction of infectious titers in lungs and clinical alleviation of the disease. Antiviral effect of favipiravir correlates with incorporation of a large number of mutations into viral genomes and decrease of viral infectivity. Antiviral efficacy is achieved with plasma drug exposure comparable with those previously found during human clinical trials. Notably, the highest dose of favipiravir tested is associated with signs of toxicity in animals. Thereby, pharmacokinetic and tolerance studies are required to determine whether similar effects can be safely achieved in humans.

[1] Unité des Virus Émergents, UVE: Aix Marseille Univ, IRD 190, INSERM 1207, Marseille, France. [2] Université de Paris, IAME, INSERM, Paris, France. [3] Laboratoire Vet-Histo, Marseille, France. [4] Laboratoire de Pharmacocinétique et Toxicologie, Hôpital La Timone, APHM, Marseille, France. [5] These authors contributed equally: Jean-Sélim Driouich, Maxime Cochin. ✉email: antoine.nougairede@univ-amu.fr

n March 2020, the World Health Organization declared coronavirus disease 2019 (COVID-19) a pandemic[1]. The COVID-19 outbreak was originally identified in Wuhan, China, in December 2019 and spread rapidly around the world within a few months. The severe acute respiratory syndrome coronavirus 2 (SARS-CoV-2), the causative agent of COVID-19, belongs to the *Coronaviridae* family and is closely related to the SARS-CoV, which emerged in China in 2002[2]. After an incubation period of about 5 days, disease onset usually begins with an influenza-like syndrome associated with high virus replication in respiratory tracts[3,4]. In some patients, a late acute respiratory distress syndrome, associated with high levels of inflammatory proteins, occurs within one to two weeks[3]. As of 11 November 2020, more than 90 million cases of COVID-19 have resulted in more than 1,936,000 deaths[5]. In the face of this ongoing pandemic and its unprecedented repercussions, not only on human health but also on society, ecology and economy, there is an urgent need for effective infection prevention and control measures.

Whilst host-directed and immune-based therapies could prove useful for the clinical management of critically ill patients, the availability of safe and effective antiviral molecules would represent an important step toward fighting the current pandemic. As conventional drug development is a slow process, repurposing of drugs already approved for any indication was extensively explored and led to the implementation of many clinical trials for the treatment of COVID-19[6]. However, the development of effective antiviral drugs for the treatment of COVID-19, should, as much as possible, rely on robust pre-clinical in vivo data, not only on efficacy generated in vitro. Accordingly, rapid implementation of rodent and non-human primate animal models should help to assess more finely the potential safety and efficacy of drug candidates and to determine appropriated dose regimens in humans[7,8].

Favipiravir (6-fluoro-3-hydroxypyrazine-2-carboxamine) is an anti-influenza drug approved in Japan that has shown broad-spectrum antiviral activity against a variety of other RNA viruses[9–15]. Favipiravir is a prodrug that is metabolized intracellularly into its active ribonucleoside 5′-triphosphate form that acts as a nucleotide analog to selectively inhibit RNA-dependent RNA polymerase and induce lethal mutagenesis[16,17]. Recently, several studies reported in vitro inhibitory activity of favipiravir against SARS-CoV-2 with 50% effective concentrations (EC$_{50}$) ranging from 62 to > 500 μM (10 to > 78 μg/mL)[18–20]. Based on these results, more than 20 clinical trials on the management of COVID-19 by favipiravir are ongoing (https://clinicaltrials.gov/).

In the present study, we evaluate the efficacy of favipiravir in vitro and using a Syrian hamster model (*Mesocricetus auratus*). Our results show that preventive or preemptive administration of high doses favipiravir induce significant reduction of infectious titers and histopathological damages in lungs and clinical alleviation of the disease. Analysis of genetic diversity of viral populations in lungs also confirms the mutagenic effect of favipiravir.

## Results

**In vitro efficacy of favipiravir.** Using VeroE6 cells and an antiviral assay based on reduction of cytopathic effect (CPE), we recorded EC$_{50}$ and EC$_{90}$ of 204 and 334 μM using a multiplicity of infection (MOI) of 0.001, 446, and > 500 μM with an MOI of 0.01 (Table 1 and Supplementary Fig. 1) in accordance with previous studies[18–20]. Infectious titer reductions (fold change in comparison with untreated cells) were ≥ 2 with 125 μM of favipiravir and ranged between 11 and 342 with 500 μM. Using Caco-2 cells, which do not exhibit CPE with SARS-CoV-2 BavPat1 strain, infectious titer reductions were around 5 with 125 μM of

favipiravir and ranged between 144 and 7721 with 500 μM of the drug. 50% cytotoxic concentrations (CC$_{50}$) in VeroE6 and Caco-2 cells were > 500 μM.

**Infection of Syrian hamsters with SARS-CoV-2.** Following Chan et al., we implemented a hamster model to study the efficacy of antiviral compounds[7]. Firstly, we intranasally infected 4-week-old female Syrian hamsters with $10^6$ TCID$_{50}$ of virus. Groups of two animals were sacrificed 2, 3, 4, and 7 days post-infection (dpi). Viral replication was quantified in sacrificed animals by RT-qPCR in organs (lungs, brain, liver, small/large bowel, kidney, spleen, and heart) and plasma. Viral loads in lungs peaked at 2 dpi, remained elevated until 4 dpi and dramatically decreased at 7 dpi (Supplementary 2). Viral loads in plasma peaked at 3 dpi and viral replication was detected in the large bowel at 2 dpi (Supplementary Fig. 2 and Supplementary Data 1). No viral RNA was detected in almost all the other samples tested (Supplementary Data 1). Subsequently, we infected animals with two lower virus inocula ($10^5$ and $10^4$ TCID$_{50}$). Viral RNA was quantified in lungs, large bowel, and plasma from sacrificed animals 2, 3, 4, and 7 dpi (Supplementary Fig. 2 and Supplementary Data 1). Viral loads in lungs peaked at 2 and 3 dpi with inocula of $10^5$ and $10^4$ TCID$_{50}$, respectively. Maximum viral loads in lungs of animals infected with each virus inoculum were comparable. Viral RNA yields in plasma and large bowel followed a similar trend but with more variability, with this two lower inocula. In addition, clinical monitoring of animals showed no marked symptoms of infection but normalized weights (i.e., % of initial weights) significantly lower from 3 dpi when compared to animals intranasally inoculated with sodium chloride 0.9% (Supplementary Fig. 2).

**In vivo efficacy of favipiravir.** To assess the efficacy of favipiravir, hamsters received the drug, intraperitoneally, three times a day (TID). We used three doses of favipiravir: 18.75, 37.5, and 75 mg/day (corresponding to 340 ± 36, 670 ± 42 and 1390 ± 126 mg/kg/day, respectively).

In a first set of experiments, treatment was initiated at the day of infection (preemptive antiviral therapy) and ended at 2 dpi. We infected groups of 6 animals intranasally with three virus inocula ($10^6$, $10^5$, and $10^4$ TCID$_{50}$) and viral replication was measured in lungs and plasma at 3 dpi (Fig. 1a). Each virus inoculum was assessed in an independent experiment. When analysis of virus replication in clarified lung homogenates was based on infectious titers (as measured using TCID$_{50}$ assay), the effect of favipiravir in reducing infectious titers was dose dependent, in particular when low virus inocula were used to infect animals (Fig. 1b). At each virus inoculum, mean infectious titers for groups of animals treated with 75 mg/day TID were significantly lower than those observed with untreated groups ($p ≤ 0.0001$): reduction of infectious titers ranged between 1.9 and 3.7 log$_{10}$. For animals infected with $10^5$ or $10^4$ TCID$_{50}$, significant infectious titer reductions of around 0.8 log$_{10}$ were also observed with the dose of 37.5 mg/day TID ($p ≤ 0.038$). Drug 90 and 99% effective doses (ED$_{90}$ and ED$_{99}$) were estimated based on these results and ranged between 31–42 mg/day and 53–70 mg/day, respectively (Table 2). When analysis of virus replication in clarified lung homogenates were assessed on viral RNA yields (as measured using quantitative real-time RT-PCR assay), significant differences with groups of untreated animals, ranging between 0.7 and 2.5 log$_{10}$, were observed only with the higher dose of favipiravir ($p ≤ 0.012$). Once again, this difference was more noticeable with lower virus inocula (Fig. 1c). Since we found higher reductions of infectious titers than those observed with viral RNA yields, we estimated the relative infectivity of viral particle (i.e., the ratio of

**Table 1 In vitro efficacy of favipiravir.**

| Cell line | MOI | Drug effective concentration[a] | | Infectious titer reduction[b] | | |
|---|---|---|---|---|---|---|
| | | $EC_{50}$ | $EC_{90}$ | 125 μM | 250 μM | 500 μM |
| Vero E6 | 0.001 | 204 μM | 334 μM | 2.2 | 13.2 | 341.9 |
| | 0.01 | 446 μM | >500 μM | 2.0 | 5.7 | 10.9 |
| Caco-2 | 0.001 | na | na | 5.6 | 137.4 | 7720.8 |
| | 0.01 | na | na | 4.0 | 7.2 | 144.0 |

*MOI* multiplicity of infection, *na* not applicable.
[a]Estimated from dose–response curves of antiviral activity (Supplementary Fig. 1).
[b]Calculated using mean infectious titers without favipiravir (virus control).

the number of infectious particles over the number of viral RNA molecules). Decreased infectivity was observed in all treated groups of animals. These differences were always significant with the higher dose of favipiravir ($p \leq 0.031$) and were significant with the dose of 37.5 mg/day TID for animals infected with $10^5$ or $10^4$ $TCID_{50}$ of virus ($p \leq 0.041$) (Fig. 1d). We then measured plasma viral loads using quantitative real-time RT-PCR assay and found, with the higher dose of favipiravir and the groups of animals infected with $10^6$ or $10^4$ $TCID_{50}$, significant reductions of 2.1 and 2.62 $log_{10}$, respectively ($p \leq 0.022$) (Fig. 1e). Finally, signs of toxicity were observed with animal treated with the dose of 75 mg/day TID: normalized weights were significantly lower than those of untreated animals (Fig. 1f).

In a second set of experiments, we assessed, over a period of 7 days, the impact of the preemptive therapy on the clinical course of the disease using weight as the primary criterion (Fig. 2a). Since signs of toxicity were noticed during the first set of experiments, we evaluated the toxicity of the three doses of favipiravir with groups of four non-infected animals treated during four days (Fig. 2b). Important toxicity was observed with the dose of 75 mg/day TID with, from the first day of treatment, normalized weights significantly lower than those of untreated animals (Supplementary Data 5). We also found a constant, but moderate, toxicity with the dose of 37.5 mg/day TID that was significant at day 4, 5, and 6 only. No toxicity was detected with the lower dose of favipiravir. To assess if the toxicity observed with the highest dose of favipiravir was exacerbated by the infection, we compared normalized weights of infected and non-infected animals treated with the dose of 75 mg/day TID. Regardless of the virus inoculum, no significant difference was observed at 1, 2, and 3 dpi (Supplementary Fig. 4). After this evaluation of favipiravir toxicity, we intranasally infected groups of 10 animals with two virus inocula ($10^5$ or $10^4$ $TCID_{50}$). Each virus inoculum was assessed in an independent experiment. Treatment with a dose of 37.5 mg/day TID was initiated on the day of infection (preemptive antiviral therapy) and ended at 3 dpi (Fig. 2a). With both virus inocula, treatment was associated with clinical alleviation of the disease (Fig. 2c, d). With the inoculum of $10^5$ $TCID_{50}$, mean weights of treated animals were significantly higher than those of untreated animals at 5 and 6 dpi ($p \leq 0.031$). Similar observations were made with the inoculum of $10^4$ $TCID_{50}$ at 5, 6, and 7 dpi ($p < 0.0001$).

In a third set of experiments, treatment was started 1 day before infection (preventive antiviral therapy) and ended at 2 dpi. We intranasally infected groups of 6 animals with $10^4$ $TCID_{50}$ of virus and viral replication was measured in lungs and plasma at 3 dpi (Fig. 3a). Once again, an inverse relationship was observed between lung infectious titers and the dose of favipiravir (Fig. 3b). Mean infectious titers for groups of animals treated with 37.5 and 75 mg/day TID were significantly lower than those observed with untreated groups ($p \leq 0.002$). Of note, undetectable infectious titers were found for all animals treated with the higher dose.

Estimated $ED_{90}$ and $ED_{99}$ were 35 and 47 mg/day, respectively (Table 2). Significant reductions of viral RNA yields of 0.9 and 3.3 $log_{10}$, were observed with animals treated with 37.5 and 75 mg/day TID, respectively ($p \leq 0.023$) (Fig. 3c). Resulting infectivity of viral particle was decreased, with a significant reduction only for the higher dose of favipiravir ($p = 0.005$) (Fig. 3e). Finally, we found significantly reduced plasma viral loads with animals treated with 37.5 and 75 mg/day TID ($p \leq 0.005$) (Fig. 3f). Once again, signs of toxicity were observed with animal treated with the dose of 75 mg/day TID: normalized weights were significantly lower than those of untreated animals (Fig. 3d).

In a last set of experiments, we assessed the impact of favipiravir treatment on lung pathological changes induced by SARS-CoV-2. Animals were intranasally infected with $10^4$ $TCID_{50}$ of virus. Treatment with two doses of favipiravir (37.5 and 75 mg/day TID) was initiated one day before infection (preventive antiviral therapy) or at day of infection (preemptive antiviral therapy) and ended at 3 dpi. For each therapeutic strategy and for each dose of favipiravir, a group of four animals was sacrificed at 3 and 5 dpi (Fig. 4a and c). As a control, we used four vehicle-treated groups of four animals (one at 3 dpi and one at 5 dpi for each therapeutic strategy). Based on the severity of inflammation, alveolar hemorrhagic necrosis and vessel lesions, a cumulative score from 0 to 10 was calculated and assigned to a grade of severity (0 = normal; 1 = mild; 2 = moderate; 3 = marked and 4 = severe; details in Supplementary Data 7). Overall, lungs of untreated animals displayed typical lesions of air-borne infection (i.e., broncho-interstitial pneumonia), with a progression between 3 dpi and 5 dpi that reflects the virus dissemination within the respiratory tree as previously demonstrated[7,21]. At 3 dpi, 7/8 untreated animals displayed mild pulmonary pathological changes (Fig. 4b and d) leading to difficulty to assess the efficacy of the treatment even if almost all mean cumulative scores of treated animals were significantly lower than those of untreated groups. In contrast, at 5 dpi all untreated animals displayed severe pulmonary impairments and we observed a dose-dependent effect of favipiravir (Fig. 4b and d). When using a preemptive antiviral strategy, all animals treated with 37.5 mg/day TID had marked histopathological damages in lungs and animals treated with 75 mg/day TID displayed mild or moderate histopathological damages (Supplementary Fig. 5). When using a preventive antiviral strategy, all animals treated with 37.5 mg/day TID had mild to marked damages in lung and animals treated with 75 mg/day TID displayed no or mild histopathological damages (Fig. 4e–h). At 5 dpi, significant cumulative score reductions were observed with both doses of favipiravir regardless the therapeutic strategy used ($p = 0.0286$, details in Supplementary Data 8).

**Favipiravir pharmacokinetics (PK) in a hamster model**. We first assessed the PK and lung distribution of favipiravir in a subgroup of uninfected animals. Groups of animals were treated

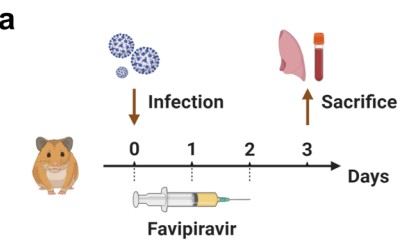

respectively with a single dose of favipiravir administrated intraperitoneally: 6.25 mg, 12.5 mg, and 25 mg. In each dose group, we sacrificed three animals at specific time points post-treatment (0.5, 1, 5 or 8 h) for determination of favipiravir in plasma. Drug concentration in lung tissue was determined at 0.5 and 5 h post-treatment. Subsequently, we assessed the favipiravir concentration after multiple dose in animals intranasally infected with $10^5$ TCID$_{50}$ of virus. Groups of nine animals received the three doses evaluated for 3 days (Fig. 1): 18.75 mg/day, 37.5 mg/day or 75 mg/day TID and were sacrificed at 12-h after the last treatment dose. Favipiravir trough concentrations were quantified in plasma ($n = 9$) and lung tissue ($n = 3$).

**Fig. 1 Virological results with preemptive favipiravir therapy. a** Experimental timeline. Groups of 6 hamsters were intranasally infected with $10^6$, $10^5$ or $10^4$ $TCID_{50}$ of virus. **b** Viral replication in lung based on infectious titers (measured using a $TCID_{50}$ assay) expressed in $TCID_{50}$/copy of ɣ-actine gene ($n = 6$ animals/group). **c** Viral replication in lung based on viral RNA yields (measured using an RT-qPCR assay) expressed in viral genome copies/copy of ɣ-actine gene ($n = 6$ animals/group). **d** Relative lung viral particle infectivities were calculated as follows: ratio of lung infectious titer over viral RNA yields ($n = 6$ animals/group). **e** Plasma viral loads (measured using an RT-qPCR assay) are expressed in viral genome copies/mL of plasma (the dotted line indicates the detection threshold of the assay) ($n = 6$ animals/group). **f** Clinical course of the disease ($n = 6$ animals/group). Normalized weight at day $n$ was calculated as follows: % of initial weight of the animal at day $n$. Data represent mean ± SD (details in Supplementary Data 2). Two-sided statistical analysis were performed using Shapiro–Wilk normality test, Student $t$-test, Mann–Whitney test, Welch's test, and two-way ANOVA with Post-hoc Dunnett's multiple comparisons test (details in Supplementary Data 3 and 4). ****, ***, ** and * symbols indicate that the average value for the group is significantly lower than that of the untreated group with a $p$-value < 0.0001, ranging between 0.0001–0.001, 0.001–0.01, and 0.01–0.05, respectively. Source data are provided as a Source data file.

**Table 2 Drug effective doses (ED) on reducing viral titers according to the level of viral inoculum.**

| Virus inoculum | ED$_{50}$ mg/day (95% CI[a]) | ED$_{90}$ mg/day (95% CI[a]) | ED$_{99}$ mg/day (95% CI[a]) |
|---|---|---|---|
| *Preemptive therapy* | | | |
| $10^4$ $TCID_{50}$ | 34 (30–37) | 42 (38–46) | 53 (48–58) |
| $10^5$ $TCID_{50}$ | 26 (21–30) | 37 (31–44) | 56 (46–65) |
| $10^6$ $TCID_{50}$ | 15 (10–20) | 31 (21–41) | 70 (48–93) |
| *Preventive therapy* | | | |
| $10^4$ $TCID_{50}$ | 27 (25–29) | 35 (32–38) | 47 (44–51) |

Dose–response curves are presented in Supplementary Fig. 3.
[a]95% confidence interval.

Results are presented in Table 3 and Supplementary Fig. 7. The single dose PK analysis showed that the maximum concentration of favipiravir was observed at 0.5 h at all doses, then plasma drug concentrations decreased exponentially to reach concentrations below 10 µg/ml at 12 h. Favipiravir PK exhibited a non-linear increase in concentration between the doses. After multiple doses, trough concentrations (12 h) of favipiravir also exhibited a non-linear increase between doses. The extrapolated 12 h post-treatment concentrations after a single dose were calculated in order to determine the accumulation ratio. Accumulation ratios were respectively 6, 16, and 21 at the three doses, confirming the non-proportional increase between doses. The average concentration after single dose administration over 0–12-h intervals was calculated and the respective values obtained were 10.1 µg/mL, 38.7 µg/mL, and 100.5 µg/mL for the three favipiravir doses.

Favipiravir lung concentrations were 1.6–2.7-fold lower than in plasma for both administration of single and multiple doses. After a single dose, the mean lung to plasma ratio ranged from 0.37 to 0.62 according to the time post-treatment and was similar between the three doses of favipiravir at 0.5 h. A high ratio 5 h post-treatment was observed at the highest dose (25 mg) with an increase by a factor 1.6–1.8 compared with the lower doses. After multiple doses, the lung penetration of favipiravir was confirmed with a mean lung to plasma ratio ranging from 0.35 to 0.44. Favipiravir was not detected in the lungs at the lowest dose (18.75 mg/day).

**Mutagenic effect of favipiravir.** To understand which genomic modifications accompanied favipiravir treatment, direct complete genome sequencing of clarified lung homogenates from animals intranasally infected with $10^6$ $TCID_{50}$ of virus and treated with the two highest doses of drug (preemptive antiviral therapy; Fig. 1) was performed. Data were generated by next-generation sequencing from lung samples of four animals per group (untreated, 37.5 mg/day TID and 75 mg/day TID). The mean sequencing coverage for each sample ranged from 10,991 to

37,991 reads per genomic position and we subjected substitutions with a frequency ≥ 1% to further analysis. The genetic variability in virus stock was also analyzed: 14 nucleotide polymorphisms were detected of which 5 recorded a mutation frequency higher than 10% (Supplementary Data 10).

In order to study the mutagenic effect of favipiravir, we used the consensus sequence from virus stock as reference and all the mutations simultaneously detected in a lung sample and in virus stock were not considered in the further analysis (1–4 mutations per sample, see Supplementary Data 10). Overall, no majority mutations were detected (mutation frequency > 50%), and almost all of the mutations occurred at a frequency lower than 10% (Fig. 5a). In addition, mutations were distributed throughout the whole genome (Fig. 5b).

Results revealed a relationship between the number of mutations detected per sample and the dose of favipiravir (Fig. 5c): the mean number of mutations increased by a factor 2 and 4.8 with groups of animals treated with 37.5 and 75 mg/day, TID respectively. The difference is significant only with a dose of 37.5 mg/day TID ($p = 0.029$). This increase of the number of mutations is mainly the consequence of the occurrence of a large number of G → A substitutions and, to a lesser extent, C → U substitutions. Consequently, regardless of the dose of favipiravir, mean frequency of G → A substitutions was significantly increased by a factor of 4.2 ($p \leq 0.009$). This rise of these transition mutations led to increased frequency of all transition mutations (significant only at dose of 37.5 mg/day TID; $p = 0.037$) and increased frequency of non-synonymous mutations (significant only at dose of 75 mg/day TID; $p = 0.009$) (Fig. 5d). We investigated whether or not effectiveness in treated animals was linked with the characteristics of the mutations detected on viral populations and found that infectious titers in lungs were negatively associated with frequency of non-synonymous and G → A mutations, and positively associated with frequency of synonymous mutations ($p < 0.03$; Fig. 5e). Finally, our experiments revealed some parallel evolution events; 32 substitutions in viral sub-populations were detected in two independent animals. Notably, 18 of these shared mutations were detected only with treated animals, 14 of them being non-synonymous (Supplementary Data 13). These mutations are located in nsp2, 3, 4, 5, 6, 14, N protein, Matrix, ORF 3a and 8. At this stage, one cannot conclude if these substitutions reflect the adaptation to the hamster model or are the result of the antiviral selection.

## Discussion

In the current study, we used a hamster model to assess efficacy of the favipiravir against SARS-CoV-2. Following infection, viral RNA was mainly detected in lungs, blood, and, to a lesser extent, in the large bowel. Peak of viral replication was observed at 2–3 dpi, in line with recently reported investigations that involved 6–10-weeks-old hamsters[7]. Clinically, the main symptom was the

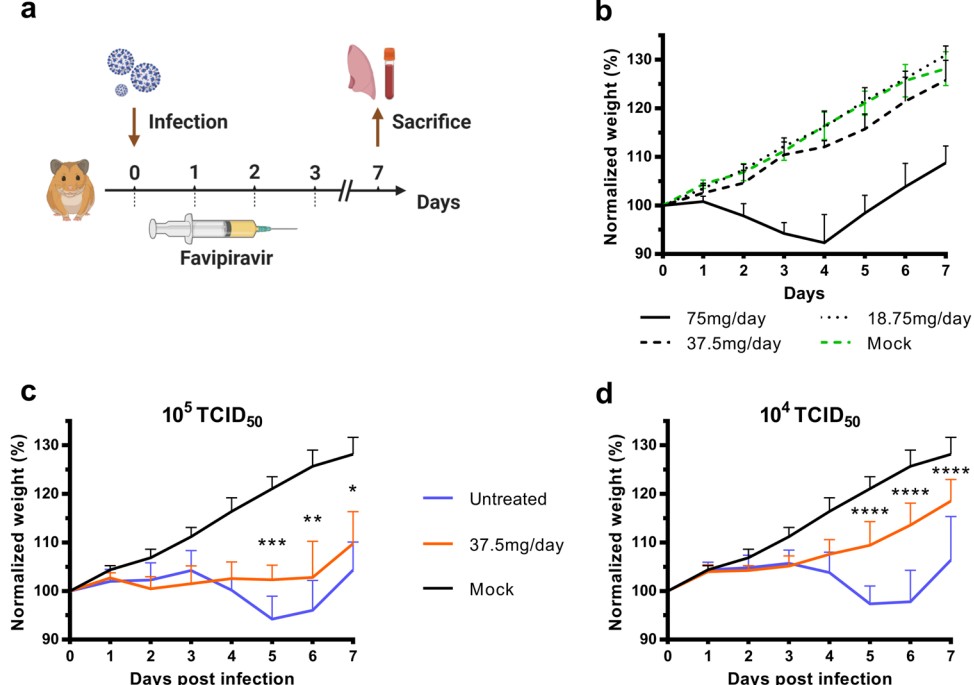

**Fig. 2 Clinical follow-up of animals. a** Experimental timeline. **b** Evaluation of the toxicity of the three doses of favipiravir (mg/day TID) with groups of four uninfected animals following the experimental timeline described in panel a but without infection. **c**, **d** Clinical follow-up with groups of 10 animals infected respectively with $10^5$ and $10^4$ TCID$_{50}$ of virus and treated with a dose of favipiravir of 37.5 mg/day TID. Normalized weight at day $n$ was calculated as follows: % of initial weight of the animal at day $n$. Data represent mean ± SD (details in Supplementary Data 2). Two-sided statistical analysis were performed using two-way ANOVA with Post-hoc Dunnett's multiple comparisons test or Post-hoc Sidak's multiple comparisons test (details in Supplementary Data 5). ****, ***, ** and * symbols indicate that the average value for the group is significantly lower than that of the untreated group with a $p$-value < 0.0001, ranging between 0.0001–0.001, 0.001–0.01, and 0.01–0.05, respectively Source data are provided as a Source data file.

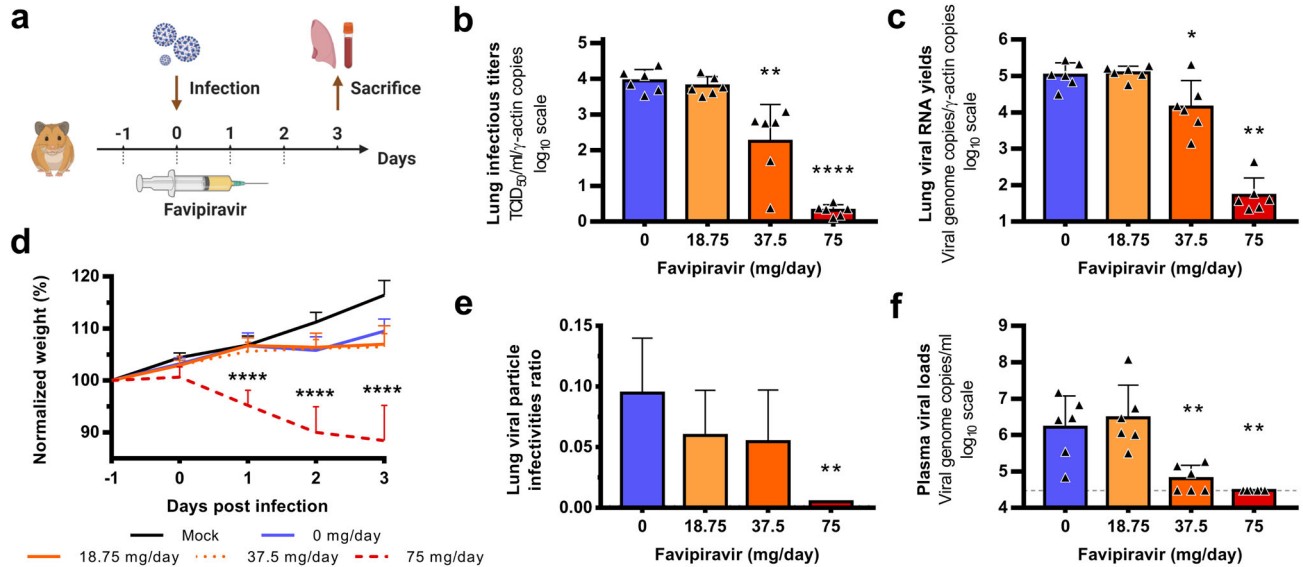

**Fig. 3 Virological results with preventive favipiravir therapy. a** Experimental timeline. Groups of 6 hamsters were intranasally infected with $10^4$ TCID$_{50}$ of virus. **b** Viral replication in lungs based on infectious titers (measured using a TCID$_{50}$ assay) expressed in TCID$_{50}$/copy of ɣ-actine gene ($n = 6$ animals/group). **c** Viral replication in lungs based on viral RNA yields (measured using an RT-qPCR assay) expressed viral genome copies/copy of ɣ-actine gene ($n = 6$ animals/group). **d** Clinical course of the disease ($n = 6$ animals/group). Normalized weight at day $n$ was calculated as follows: % of initial weight of the animal at day $n$. **e** Relative lung virus infectivities were calculated as follows: ratio of lung infectious titer over viral RNA yields ($n = 6$ animals/group). **f** Plasma viral loads (measured using an RT-qPCR assay) are expressed in viral genome copies/mL of plasma (the dotted line indicates the detection threshold of the assay) ($n = 6$ animals/group). Data represent mean ± SD (details in Supplementary Data 2). Statistical analysis were performed using Shapiro–Wilk normality test, Student $t$-test, Mann–Whitney test, One-sample $t$-test and two-way ANOVA with Post-hoc Dunnett's multiple comparisons test (details in Supplementary Data 3 and 4). ****, ** and * symbols indicate that the average value for the group is significantly different from that of the untreated group with a $p$-value < 0.0001, ranging between 0.001–0.01 and 0.01–0.05, respectively. Source data are provided as a Source data file.

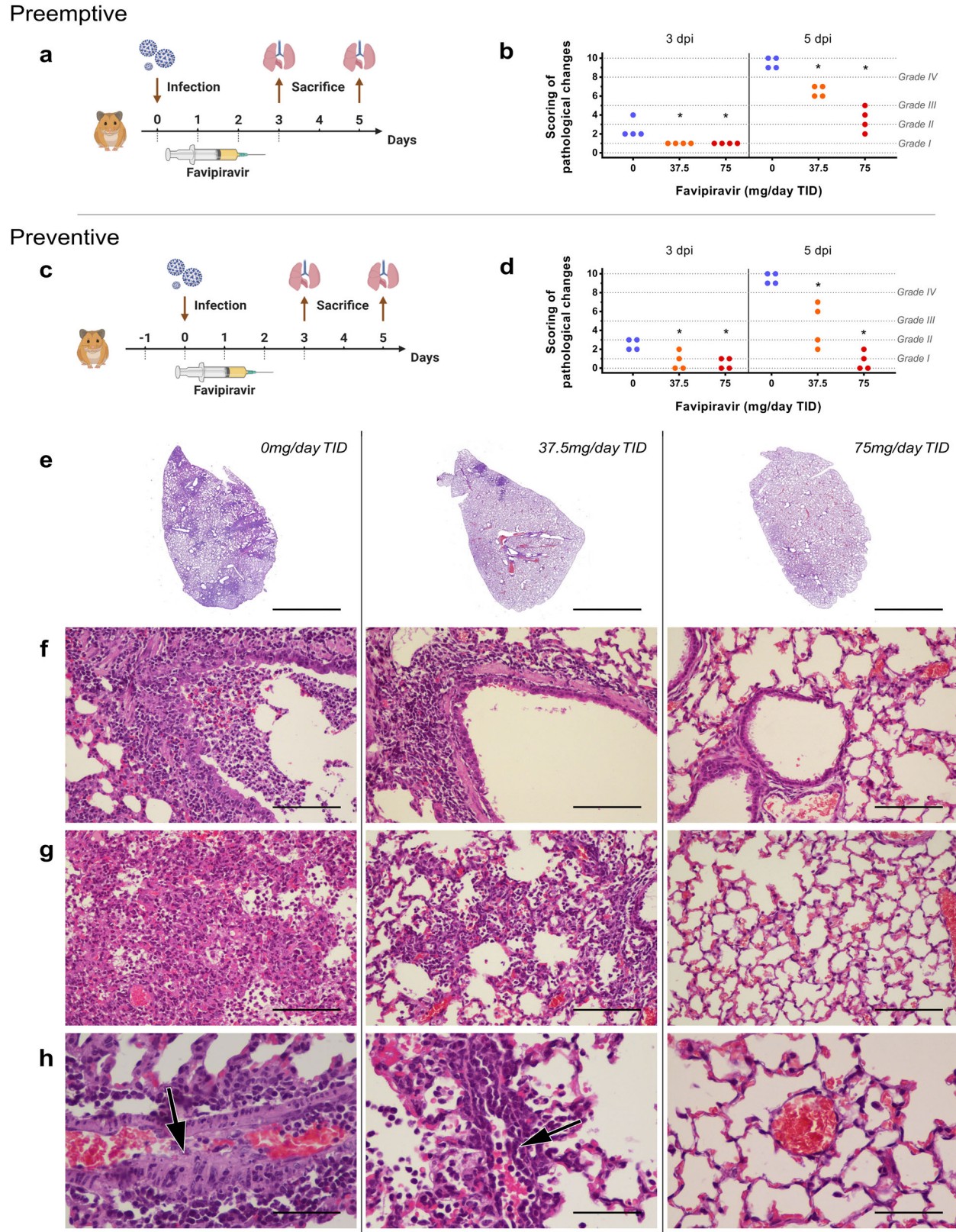

lack of weight gain, observed from the first day of infection and followed by recovery at 7 dpi. Histopathological changes are comparable to those previously described[7,21]. Notably, our results revealed that all animals with marked or severe pulmonary impairments displayed vascular lesions (endothelitis, vasculitis) as previously described in humans[22]. Overall, this confirmed that

the in vivo model, with younger animals (4 weeks-old), is suitable for preclinical evaluation of antiviral compounds against SARS-CoV-2.

Using a preemptive strategy, we demonstrated that doses of favipiravir of around 700–1400 mg/kg/day TID reduced viral replication in the lungs of infected animals and allowed clinical

**Fig. 4 Lung histopathological changes with preemptive or preventive favipiravir therapy.** Groups of four animals were intranasally infected with $10^4$ $TCID_{50}$ of virus and sacrificed at 3 and 5 dpi. Experimental timelines for preemptive (**a**) and preventive (**c**) favipiravir therapies. At day of sacrifice, lungs were collected, fixed, and embedded in paraffin. Tissue sections were stained with hematoxylin-eosin (H&E). Based on severity of inflammation, alveolar hemorrhagic necrosis, and vessel lesions, a cumulative score from 0 to 10 was calculated and assigned to a grade of severity (I, II, III, and IV). Scoring of pathological changes for preemptive (**b**) and preventive (**d**) favipiravir therapies ($n = 4$ animals/group) (details in Supplementary Data 7). Two-sided statistical analysis were performed using Shapiro–Wilk normality test, Student t-test, Mann–Whitney test, and two-way ANOVA with Post-hoc Dunnett's multiple comparisons test (details in Supplementary Data 7 and 8). * Symbol indicates that the average value for the group is significantly different from that of the untreated group with a p-value ranging between 0.01 and 0.05. **e** Representative images of lung tissue (left lobe) (scale bar: 4 mm): multifocal and extensive areas of inflammation for untreated animal, multifocal but limited areas of inflammation for 37.5 mg/day treated animal and normal lung for 75 mg/day treated animal ($n = 4$ samples/group). **f** Representative images of bronchial inflammation (scale bar: 100 µ): severe peribronchiolar inflammation and bronchiole filled with neutrophilic exudates for untreated animal, mild peribronchiolar inflammation for 37.5 mg/day treated animal and normal bronchi for 75 mg/day treated animal ($n = 4$ samples/group). **g** Representative images of alveolar inflammation (scale bar: 100 µ): severe infiltration of alveolar walls, alveoli filled with neutrophils/macrophages for untreated animal, moderate infiltration of alveolar walls, some alveoli filled with neutrophils/macrophages for 37.5 mg/day treated animal and normal alveoli for 75 mg/day treated animal. **h** Representative images of vessel inflammation (scale bar: 50 µ): infiltration of vascular wall with neutrophils/cell debris and endothelial damage (arrow) for untreated animal, moderate endothelial leukocytic accumulation for 37.5 mg/day treated animal and normal vessel for 75 mg/day treated animal ($n = 4$ samples/group). Clinical courses of the disease are presented in Supplementary Fig. 6. Source data are provided as a Source data file.

**Table 3 Plasma and lung concentrations of favipiravir after administration of a single dose or multiple dose of favipiravir.**

| Time post-treatment | Single dose | | | | Multiple dose[a] (Day 3) | | | |
|---|---|---|---|---|---|---|---|---|
| | Dose | Plasma (µg/mL) | Lung (µg/g) | L/p ratio | Dose | Plasma (µg/mL) | Lung (µg/g) | L/p ratio |
| 0.5 h | 25 mg | 372 ± 47.5 | 216 ± 39 | 0.58 ± 0,04 | 75 mg/day TID | | | |
| 1 h | | 279 ± 49.9 | | | | | | |
| 5 h | | 135 ± 49.0 | 81.3 ± 24 | 0.62 ± 0.10 | | | | |
| 8 h | | 5.77 ± 1.34 | | | | | | |
| 12 h | | 1.43[b] | | | | 29.9 ± 9.83 | 16.0 ± 4.87 | 0.44 ± 0,07 |
| 0.5 h | 12.5 mg | 166 ± 52.0 | 90.7 ± 12.7 | 0.58 ± 0.14 | 37.5 mg/day TID | | | |
| 1 h | | 155 ± 20.6 | | | | | | |
| 5 h | | 10.7 ± 5.16 | 3.84 ± 1.49 | 0.37 ± 0.052 | | | | |
| 8 h | | 1.94 ± 0.06 | | | | | | |
| 12 h | | 0.16[b] | | | | 2.57 ± 1.22 | 1.36 ± 0.14 | 0.35 ± 0,03 |
| 0.5 h | 6.25 mg | 86.3 ± 4.11 | 50.2 ± 16.4 | 0.58 ± 0.17 | 18.75 mg/day TID | | | |
| 1 h | | 35.2 ± 27.8 | | | | | | |
| 5 h | | 2.90 ± 0.25 | 1.09 ± 0.05 | 0.38 ± 0.05 | | | | |
| 8 h | | 0.56 ± 0.16 | | | | | | |
| 12 h | | 0.05[b] | | | | 0.31 ± 0.14 | Not detected | NA |

NA not applicable.
Data represent mean ± SD; three animals for each condition except at multiple dose ($n = 9$ for plasma; $n = 3$ for lung); details in Supplementary Data 9.
[a]PK realized after 3 days of favipiravir administered three times a day, at the end of the dosing interval (trough concentrations).
[b]Extrapolated $C_{12h}$.

alleviation of the disease (Figs. 1 and 2). In the most favorable situation, where high doses were used as a preventive therapy, favipiravir led to undetectable viral replication in lung and plasma. These results showed that the use of high doses of favipiravir could expand its in vivo spectrum against RNA viruses. Reduction of viral replication was greater when estimated on the basis of infectious titers than on total viral RNA as previously observed in non-human primates treated with Remdesivir and in hamsters treated with favipiravir[23,24]. Furthermore, the analysis of pulmonary histopathological changes revealed that favipiravir played a protective role by reducing the severity of the lesions. However, the effective doses of favipiravir were higher than those usually used in rodent models (≈100–400 mg/kg/day)[10,12,25–28]. This can be correlated with the high favipiravir $EC_{50}$ found in vitro for SARS-CoV-2. Moreover, effective doses were associated with significant toxicity in our hamster model (Fig. 2). This observed toxicity reflected only the adverse effects of favipiravir and was not exacerbated during SARS-CoV-2 infection. Indeed, similar weights were measured among infected and non-infected animals treated with the highest dose of favipiravir at 1, 2, and 3 dpi.

In the present study, reduction of viral replication was correlated with the dose of favipiravir administrated and inversely correlated with the virus inoculum. In a recent study, the efficacy of favipiravir intraperitoneally or orally administered twice daily (loading dose of 900 and 1200 mg/kg/day followed by 600 and 1000 mg/kg/day, respectively) was assessed using a similar hamster model (6–10 weeks old) with high virus inocula ($2 \times 10^6$ $TCID_{50}$)[24]. Treatment with the highest dose of favipiravir resulted in a moderate decrease of viral RNA yields in lung tissue and the lowest dose induced an even smaller inhibitory effect. However, significant infectious titers reduction were observed in a dose-dependent manner in lungs. Both doses were also associated with regression of pulmonary histopathological impairments. Overall, these results are in accordance with ours at the medium and the high doses of favipiravir (around 670 and 1390 mg/kg/day TID). However, in this other study, no signs of toxicity were associated with favipiravir treatment regardless the dosing regimen. This discrepancy could be due to the difference between (i) the highest daily doses used (1000 mg/kg/day in regards to 1390 mg/kg/day in our study), (ii) the dosing regimens (BID instead of TID in our

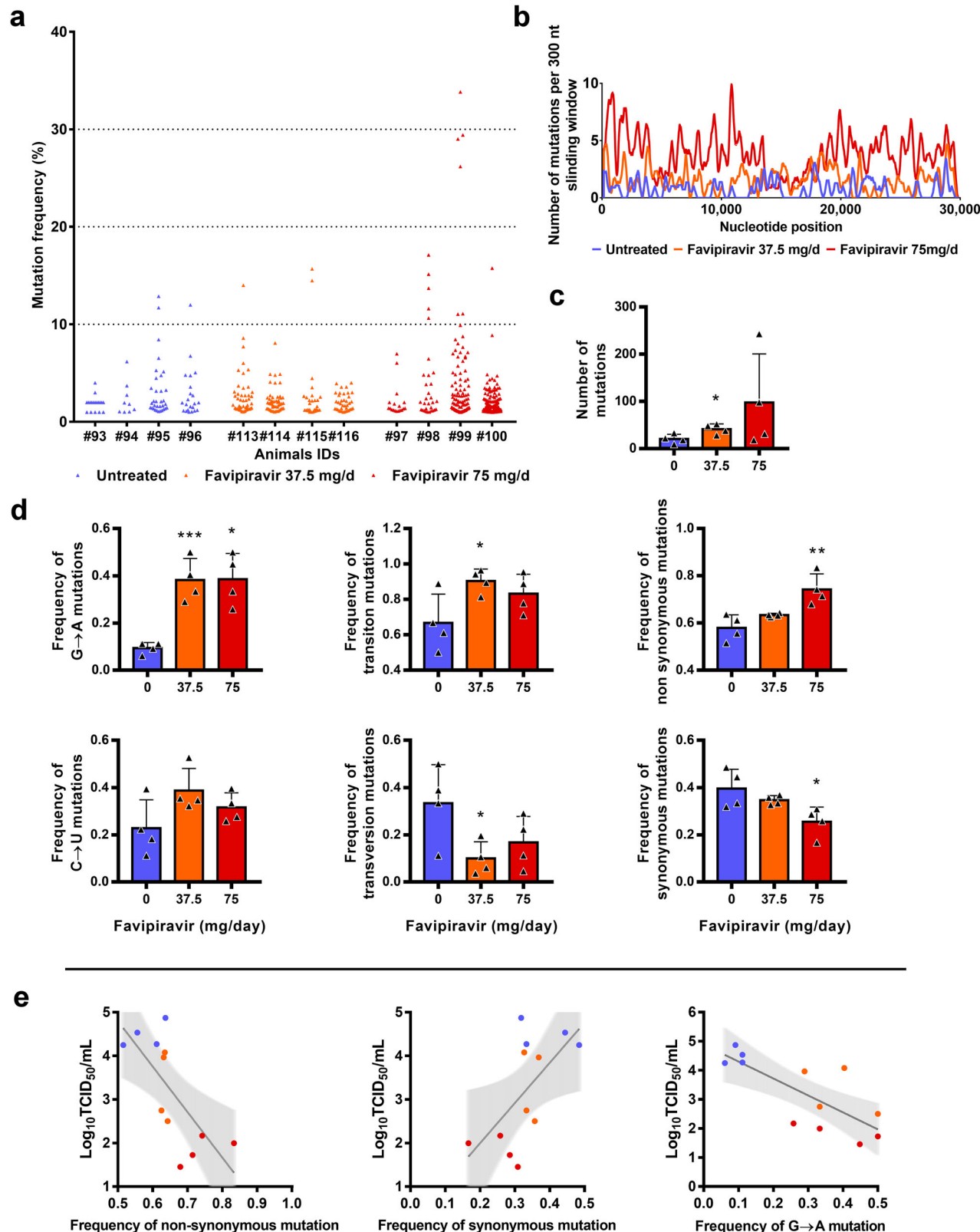

study), and/or (iii) the age of the hamsters at day of infection (6–10 weeks old in comparison to 4 weeks old in our study).

With influenza viruses, favipiravir acts as a nucleotide analog since it is recognized as a purine nucleotide by the viral RNA-dependent RNA polymerase. It is metabolized intracellularly to its active form and incorporated into nascent viral RNA strands.

This inhibits RNA strand extension and induces abnormal levels of mutation accumulation into the viral genome[16,17]. Recently, it was shown in vitro that favipiravir has a similar mechanism of action with SARS-CoV-2 through a combination of chain termination, reduced RNA synthesis and lethal mutagenesis[20]. Our genomic analysis confirmed the mutagenic effect of favipiravir

**Fig. 5 Mutagenic effect of favipiravir. a** Viral genetic diversity in clarified lung homogenates. For each condition, four samples were analyzed. Each triangle represents a mutation (only substitutions with a frequency ≥ 1% were considered). **b** Patterns of mutation distribution on complete viral genome. Each variable nucleotide position was counted only once when found. The variability was represented using 75 nt sliding windows. For each condition, variable nucleotide positions were determined and represented using a 300 nt sliding window. **c** Mean number of mutations ($n = 4$ samples/group). Data represent mean ± SD. **d** Mutation characteristics ($n = 4$ samples/group). For each sample, the frequency of a given mutation was calculated as follows: number of this kind of mutation detected in the sample divided by the total number of mutations detected in this sample. Data represent mean ± SD (details in details in Supplementary Data 10 and 13). Two-sided statistical analysis were performed using Shapiro–Wilk normality test, Student $t$-test, Mann–Whitney test, and Welch's test (details in Supplementary Data 11 and 12). ***, ** and * symbols indicate that the average value for the group is significantly lower than that of the untreated group with a p-value ranging between 0.0001–0.001, 0.001–0.01, and 0.01–0.05, respectively. **e** Association between lung infectious titers (measured using a $TCID_{50}$ assay) and frequency of non synonymous, synonymous and $G \rightarrow A$ mutations. Each dot represent data from a given animal. Statistical analysis was performed using univariate linear regression. The error band (in gray) represent the 95% confidence interval of the regression line. Source data are provided as a Source data file.

in vivo[24]. Indeed, we found that favipiravir treatment induced appearance of a large number of $G \rightarrow A$ mutations into viral genomes (Fig. 5). This was associated to a decrease of viral infectivity probably because alteration of the genomic RNA disturb the replication capacity. Similar findings were described in vitro and in vivo with other RNA viruses[9,16,29,30]. Of note, we also observed a strong inverse association between infectious titers in lungs and the proportion of non-synonymous mutations detected in viral populations. Because random non-synonymous mutations are more deleterious than synonymous mutations[31], this suggests that they were randomly distributed over the three positions of the codons and that no compensatory mechanism was triggered by the virus to eliminate them (i.e. negative selection). Finally, the inverse correlation between lung infections titers and the frequency of $G \rightarrow A$ substitutions showed that an increased proportion of these mutations beyond an error threshold might be expected to cause lethal mutagenesis.

Genomic analyses revealed that 18 mutations detected in viral sub-populations were shared only with treated animals. Two of them were located in the nsp14 coding region involved in the proof-reading activity of the viral RNA polymerisation[32,33]. However, they were located in the N7 MTase domain involved in viral RNA capping[34,35]. By comparison, resistance mutations selected against Remdesivir in β-coronavirus murine hepatitis virus model were obtained in the RdRP (nsp12) coding sequence[36]. Further investigations are needed to assess the impact of these mutations on the antiviral effect of favipiravir.

Favipiravir PK in our hamster model displayed a non-linear increase in plasma exposure between the doses as already reported in non-human primates[37]. The observed favipiravir concentration versus time profiles were in agreement with previous results of a PK study performed in 7–8-week-old hamsters orally treated with a single dose of 100 mg/kg of favipiravir[38]. The maximum plasma drug concentration occurred at 0.5 h after oral administration, earlier than in humans, and then decreased rapidly in agreement with its short half-life[39]. After repeated doses, plasma exposure confirmed non-linear PK over the entire range of doses, further emphasized by accumulation ratios. The important accumulation observed at the highest dose could explain in part the toxicity observed in hamsters at this dose. Favipiravir undergoes an important hepatic metabolism mainly by aldehyde oxidase producing an inactive M1 metabolite and inhibits aldehyde oxidase activity in a concentration- and time-dependent manner. These properties explain the self-inhibition of its own metabolism as observed in our study in which the highest dose of favipiravir led to a greater increase in favipiravir concentrations[40].

A good penetration of favipiravir in lungs was observed with lung/plasma ratios ranging from 35 to 44% after repeated doses, consistent with its physicochemical properties. Lung exposure was also in accordance with the previous studies[38].

The medium dose of favipiravir used in this study (670 mg/kg/day TID) is within the range of the estimated doses required to reduce by 90% ($ED_{90}$) the level of infectious titers in lungs (ranging between 31 and 42 mg/day corresponding to 570–780 mg/kg/day) (Table 2) and displayed limited drug-associated toxicity (Fig. 2b). Animals infected with $10^5$ and $10^4$ $TCID_{50}$ of virus, and treated following a preemptive strategy with this dose displayed significant reduction of infectious titers and histopathological scores in lungs and clinical alleviation of the disease (Figs. 1, 2, and 4). Animal treated following a preventive strategy with this dose also displayed significant reduction of viral replication and histopathological scores in lungs (Figs. 3 and 4). Regarding the accumulation ratio after repeated doses and the good penetration of favipiravir in lungs, effective concentrations can be expected in lungs, throughout the course of treatment using this dose of 670 mg/kg/day TID.

How clinically realistic are these results? To address this question we compared the drug concentrations obtained in the hamster model with those obtained in patients. In 2016, a clinical trial evaluated the use of favipiravir in Ebola-infected patients[41]. The dose used in Ebola-infected patients was 6000 mg on day 0 followed by 1200 mg BID for 9 days. The median trough concentrations of favipiravir at day 2 and day 4 were 46.1 and 25.9 μg/mL, respectively. This is within the range observed here in hamsters treated with the highest dose (around 1400 mg/kg/day), with a mean trough concentration of 29.9 μg/mL. However, additional investigations are required to determine whether or not similar favipiravir plasma exposure in SARS-COV-2 infected patients are associated with antiviral activity. The major differences in PK between hamster and humans, and the toxicity observed at the highest doses in our animal model limits the extrapolation of our results. Therefore, whether safe dosing regimens in humans may achieve similar plasma exposure and recapitulate the profound effect on viral replication is unknown. Further, the intracellular concentration of the active metabolite was not determined and which parameter of the drug pharmacokinetics best drives the antiviral effect remains to be established.

In summary, this study establishes that high doses of favipiravir are associated with antiviral activity against SARS-CoV-2 infection in a hamster model. The better antiviral efficacy was observed using a preventive strategy, suggesting that favipiravir could be more appropriate for a prophylactic use. Our results should be interpreted with caution because high doses of favipiravir were associated with signs of toxicity in our model. It is required to determine if a tolerable dosing regimen could generate similar exposure in non-human primates, associated with significant antiviral activity, before testing a high dose regimen in COVID-19 patients. Furthermore, subsequent studies should determine if an increased antiviral efficacy can be reached using favipiravir in association with other effective antiviral drugs, since

this strategy may enable to reduce the dosing regimen of favipiravir. Finally, this work reinforces the need for rapid development of animal models to confirm in vivo efficacy of antiviral compounds and accordingly, to determine appropriate dose regimens in humans before starting clinical trials.

## Methods

**Cells**. VeroE6 cells (ATCC CRL-1586) and Caco-2 cells (ATCC HTB-37) were grown at 37 °C with 5% $CO_2$ in minimal essential medium (MEM) supplemented with 7.5% heat-inactivated fetal bovine serum (FBS), 1% penicillin/streptomycin and 1% non-essential amino acids (all from ThermoFisher Scientific).

**Virus**. All experiments with infectious virus were conducted in biosafety level (BSL) 3 laboratory. SARS-CoV-2 strain BavPat1, supplied through European Virus Archive GLOBAL (https://www.european-virus-archive.com/), was provided by Christian Drosten (Berlin, Germany). Virus stocks were prepared by inoculating at MOI of 0.001 a 25 cm 2 culture flask of confluent VeroE6 cells with MEM medium supplemented with 2.5% FBS. The cell supernatant medium was replaced each 24 h and harvested at the peak of infection, supplemented with 25 mM HEPES (Sigma), aliquoted and stored at −80 °C.

**In vitro determination of $EC_{50}$, $EC_{90}$, $CC_{50}$, and infectious titer reductions**. One day prior to infection, $5 \times 10^4$ VeroE6 cells were seeded in 96-well culture plates ($5 \times 10^4$ cells/well in 100 μL of 2.5% FBS medium (assay medium). The next day, seven 2-fold serial dilutions of favipiravir (Courtesy of Toyama-Chemical; 0.61 μg/mL to 78.5 μg/mL, in triplicate) were added (25 μL/well, in assay medium). Eight virus control wells were supplemented with 25 μL of assay medium and eight cell controls were supplemented with 50 μL of assay medium. After 15 min, 25 μL of virus suspension, diluted in assay medium, was added to the wells at an MOI of 0.01 or 0.001 (except for cell controls). Three days after infection, cell supernatant media were collected to perform $TCID_{50}$ assay (at concentration of 500, 250, and 125 μM), as described below, in order to calculate infectious titer reductions and cell viability was assessed using CellTiter-Blue reagent (Promega) following the manufacturer's instructions. Fluorescence (560/590 nm) was recorded with a Tecan Infinite 200Pro machine (Tecan). The 50 and 90% effective concentrations ($EC_{50}$, $EC_{90}$) were determined using logarithmic interpolation (% of inhibition were calculated as follows: $(OD_{sample} − OD_{virus\ control})/(OD_{cell\ control} − OD_{virus\ control})$). For the evaluation of $CC_{50}$ (the concentration that induced 50% cytoxicity), the same culture conditions were set as for the determination of the $EC_{50}$, without addition of the virus, then cell viability was measured using CellTiter Blue (Promega). $CC_{50}$ was determined using logarithmic interpolation.

### In vivo experiments

*Approval and authorization*. In vivo experiments were approved by the local ethical committee (C2EA—14) and the French 'Ministère de l'Enseignement Supérieur, de la Recherche et de l'Innovation' (APAFIS#23975) and performed in accordance with the French national guidelines and the European legislation covering the use of animals for scientific purposes. All experiments were conducted in BSL 3 laboratory.

*Animal handling*. Three-week-old female Syrian hamsters were provided by Janvier Labs. Animals were maintained in ISOcage P - Bioexclusion System (Techniplast) with unlimited access to water/food and 14 h/10 h light/dark cycle. Animals were weighed and monitored daily for the duration of the study to detect the appearance of any clinical signs of illness/suffering. Virus inoculation was performed under general anesthesia (isoflurane). Organs and blood were collected after euthanasia (cervical dislocation) which was also realized under general anesthesia (isoflurane).

*Hamster Infection*. Anesthetized animals (four-week-old) were intranasally infected with 50 μL containing $10^6$, $10^5$ or $10^4$ $TCID_{50}$ of virus in 0.9% sodium chloride solution. The mock group was intranasally inoculated with 50 μL of 0.9% sodium chloride solution.

*Favipiravir administration*. Hamster were intraperitoneally inoculated with different doses of favipiravir. Control group were intraperitoneally inoculated with a 0.9% sodium chloride solution.

*Organ collection*. Organs were first washed in 10 mL of 0.9% sodium chloride solution and then transferred to a 2 mL or 50 mL tube containing respectively 1 mL (small/large bowel pieces, kidney, spleen, and heart) or 10 mL (lungs, brain and liver) of 0.9% sodium chloride solution and 3 mm glass beads. They were crushed using a Tissue Lyser machine (Retsch MM400) for 5 min at 30 cycles/s and then centrifuged 5 min at $16,200 \times g$. Supernatant media were transferred to a 2 mL tube, centrifuged 10 min at $16,200 \times g$, and stored at −80 °C. One milliliter of blood was harvested in a 2 mL tube containing 100 μL of 0.5 M EDTA (ThermoFischer Scientific). Blood was centrifuged for 10 min at $16,200 \times g$ and stored at −80 °C.

**Quantitative real-time RT-PCR (RT-qPCR) assays**. To avoid contamination, all experiments were conducted in a molecular biology laboratory that is specifically designed for clinical diagnosis using molecular techniques, and which includes separate laboratories dedicated to perform each step of the procedure. Prior to PCR amplification, RNA extraction was performed using the QIAamp 96 DNA kit, and the Qiacube HT kit and the Qiacube HT (both from Qiagen) following the manufacturer's instructions. Shortly, 100 μl of organ clarified homogenates, spiked with 10 μL of internal control (bacteriophage MS2)[42], were transferred into an S-block containing the recommended volumes of VXL, proteinase K, and RNA carrier. RT-qPCR (SARS-CoV-2 and MS2 viral genome detection) were performed using the Express one step RT-qPCR Universal kit (ThermoFisher Scientific) using 3.5 μL of RNA and 6.5 μL of RT-qPCR mix that contains 250 nmol of each primer and 75 nmol of probe. Amplification was performed with the QuantStudio 12K Flex Real-Time PCR System (ThermoFisher Scientific) using the following conditions: 50 °C for 10 min, 95 °C for 20 s, followed by 40 cycles of 95 °C for 3 s, 60 °C for 30 s. qPCR (ɣ-actine gene detection) was perfomed under the same condition as RT-qPCR with the following modifications: we used the Express one step qPCR Universal kit (ThermoFisher Scientific) and the 50 °C step of the amplification cycle was removed. Data were collected using the QuantStudio 12K Flex Software v1.2.3. Primers and probes sequences used to detect SARS-CoV-2, MS2 and ɣ-actine are described in Supplementary Table 1.

**Tissue-culture infectious dose 50 ($TCID_{50}$) assay**. To determine infectious titers, 96-well culture plates containing confluent VeroE6 cells were inoculated with 150 μL per well of serial dilutions of each sample (four-fold or ten-fold dilutions when analyzing lung clarified homogenates or cell supernatant media, respectively). Each dilution was performed in sextuplicate. Plates were incubated for 4 days and then read for the absence or presence of cytopathic effect in each well. Infectious titers were estimated using the method described by Reed & Muench[43].

**Favipiravir pharmacokinetics**. Animal handling, hamster infections, and favipiravir administrations were performed as described above. A piece of left lung was first washed in 10 mL of sodium chloride 0.9% solution, blotted with filter paper, weighed and then transferred to a 2 mL tube containing 1 mL of 0.9% sodium chloride solution and 3 mm glass beads. It was crushed using the Tissue Lyser machine (Retsch MM400) during 10 min at 30 cycles/s and then centrifuged 5 min at $16,200 \times g$. Supernatant media were transferred to 2 mL tubes, centrifuged 10 min at $16,200 \times g$ and stored at −80 °C. One milliliter of blood was harvested in a 2 mL tube containing 100 μL of 0.5 M EDTA (ThermoFischer Scientific). Blood was centrifuged for 10 min at $16,200 \times g$ and stored at −80 °C.

Quantification of favipiravir in plasma and lung tissues was performed by a validated sensitive and selective validated high-performance liquid chromatography coupled with tandem mass spectrometry method (UPLC-TQD, Waters, USA) with a lower limit of quantification of 0.1 μg/mL. Precision and accuracy of the three quality control samples (QCs) were within 15% over the calibration range (0.5 μg/mL to 100 μg/mL) (Bekegnran et al., submitted). Favipiravir was extracted by a simple protein precipitation method, using acetonitrile for plasma and ice-cold acetonitrile for clarified lung homogenates. Briefly, 50 μL of samples matrix was added to 500 μL of acetonitrile solution containing the internal standard (favipiravir-13C,15N, Alsachim), then vortexed for 2 min followed by centrifugation for 10 min at 4 °C. The supernatant medium was evaporated and the dry residues were then transferred to 96-well plates and 50 μL was injected. To assess the selectivity and specificity of the method and matrix effect, blank plasma and tissues homogenates from 2 control animals (uninfected and untreated) were processed at each run. Moreover, the same control samples spiked with favipiravir concentration equivalent to the QCs (0.75, 50, and 80 μg/mL) were also processed and compared to the QCs samples. Data were collected using the MassLynx Mass Spectrometry Software 4.1.

Noncompartemental analysis conducted using software Pkanalix2019R2 (www.lixoft.com). Areas under the plasma concentration time curve were computed using medians of favipiravir concentrations at 0.5, 1, 5, and 8 h, and extrapolated until $T = 12$ h. $C_{trough}$ were extrapolated at $T = 12$ h using lambda-z loglinear regression on the decreasing slope of concentrations.

**Histology**. Animal handling, hamster infections, and favipiravir administrations were performed as described above. Lungs were collected after intratracheal instillation of 4% (w/v) formaldehyde solution, fixed 72 h at room temperature with a 4% (w/v) formaldehyde solution and embedded in paraffin. Tissue sections of 3.5 μm, obtained following guidelines from the "global open RENI" (The standard reference for nomenclature and diagnostic criteria in toxicologic pathology; https://www.goreni.org/), were stained with hematoxylin-eosin (H&E) and blindly analyzed by a certified veterinary pathologist. Microscopic examination was done using a Nikon Eclipse E400 microscope. Different anatomic compartments were examined (see Supplementary Table 2): (1) for bronchial and alveolar walls, a score of 0 to 4 was assigned based on the severity of inflammation; (2) regarding alveoli, a score of 0 to 2 was assigned based on presence and severity of hemorrhagic necrosis; (3) regarding vessel lesions (endothelitis/vasculitis), absence or presence was scored 0 or 1 respectively. A cumulative score was then calculated and assigned to a grade of severity (see Supplementary Table 3).

**Sequence analysis of the full-length genome**. 200 μL of lung clarified homogenate or infectious cell supernatant (virus stock) was inactivated with an equal volume of VXL lysis buffer (Qiagen) and viral RNA was extracted using an EZ1 Advanced XL robot with the EZ1 mini virus 2.0 kit (both from Qiagen) and linear acrylamide (ThermoFisher Scientific) in place of carrier RNA. cDNA was generated in a final volume of 40 μL using 14 μL of nucleic acid extract, random hexamer and the Protoscript II First Strand cDNA Synthesis Kit (New England Biolabs). A specific set of primers (Supplementary Table 4) was used to generate thirteen amplicons covering the entire genome with the Q5 High-Fidelity DNA polymerase (New England Biolabs). PCR mixes (final volume 25 μL) contained 2.5 μL of cDNA, 2 μL of each primer (10 μM), and 12.5 μL of Q5 High-Fidelity 2X Master Mix. Amplification was performed with the following conditions: 30 s at 98 °C, then 45 cycles of 15 s at 98 °C and 5 min at 65 °C. Size of PCR products was verified by gel electrophoresis. For each sample, an equimolar pool of all amplicons was prepared and purified using Monarch PCR & DNA Cleanup Kit (New England Biolabs). After DNA quantification using Qubit dsDNA HS Assay Kit and Qubit 2.0 fluorometer (ThermoFisher Scientific), amplicons were fragmented by sonication into fragments of around 200 bp long. Libraries were built by adding barcodes, for sample identification, and primers using AB Library Builder System (ThermoFisher Scientific). To pool equimolarly the barcoded samples a quantification step by real-time PCR using Ion Library TaqMan Quantitation Kit (ThermoFisher Scientific) was performed. Then, emulsion PCR from pools and loading on 530 chip was performed using the automated Ion Chef instrument (ThermoFisher Scientific). Sequencing was performed using the S5 Ion torrent technology v5.12 (ThermoFisher Scientific) following the manufacturer's instructions. Consensus sequence was obtained after trimming of reads (reads with quality score < 0.99, and length < 100 pb were removed and the 30 first and 30 last nucleotides were removed from the reads). Mapping of the reads on a reference (determine following blast of De Novo contigs) was done using CLC genomics workbench software v.20 (Qiagen). A de novo contig was also produced to ensure that the consensus sequence was not affected by the reference sequence. Mutation frequency for each position was calculated as the number of reads with a mutation compared to the reference divided by the total number of reads at that site. Only substitutions with a frequency of at least 1% were taken into account for the analysis (Supplementary Data 10).

**ED50, ED90, and ED99 determination**. We conducted a nonlinear regression of infectious viral load against dose, using an $E_{max}$ model, giving $VL = VL_0 \times \left(1 - \left(\frac{D^\gamma}{D^\gamma + D_{50}^\gamma}\right)\right)$ with $VL_0$ being infectious viral load of untreated animals. We estimated $D_{50}$ the dose required to decrease viral load by 50%, using a coefficient $\gamma$ to account for the high sigmoidicity of the relation between dose and titers. $\gamma$ coefficient was chosen as the one maximizing likelihood of the model. We extrapolated the $D_{90}$ and $D_{99}$ using $D_{90} = \sqrt[\gamma]{9 \times D_{50}^\gamma}$ and $D_{99} = \sqrt[\gamma]{99 \times D_{50}^\gamma}$, as well as their 95% confidence interval using the delta method.

**Graphical representations and statistical analysis**. Graphical representations and statistical analyses were performed with Graphpad Prism 7 (Graphpad software) except linear/nonlinear regressions and their corresponding graphical representations that were performed using R statistical software (http://www.R-project.org). Statistical details for each experiment are described in the figure legends and in corresponding Supplementary data. When relevant, two-sided statistical tests were always used. P-values lower than 0.05 were considered statistically significant. Experimental timelines were created on biorender.com.

**Reporting summary**. Further information on research design is available in the Nature Research Reporting Summary linked to this article.

## Data availability
Raw sequence reads of the virus genome analyzed in this study have been deposited in the BioProject data bank (PRJNA648821). Authors can confirm that all other relevant data are included in the paper and/or its Supplementary information files. Source data are provided with this paper.

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

## Acknowledgements

We thank Laurence Thirion (UVE; Marseille) for providing RT-qPCR systems. We thank Camille Placidi (UVE; Marseille) for her technical contribution. We thank Lionel Chasson (CIML; Marseille) for helping to generate low magnification pictures. We also thank Pr. Ernest A. Gould (UVE; Marseille) for his careful reading of the manuscript and English language editing. We thank Pr. Drosten and Pr. Drexler for providing the SARS-CoV-2 strain through the European Research infrastructure EVA GLOBAL. This work was supported by the Fondation de France "call FLASH COVID-19", project TAMAC, by "Institut national de la santé et de la recherche médicale" through the REACTing (REsearch and ACTion targeting emerging infectious diseases) initiative ("Preuve de concept pour la production rapide de virus recombinant SARS-CoV-2"), and by European Virus Archive Global (EVA 213 GLOBAL) funded by the European Union's Horizon 2020 research and innovation program under grant agreement No. 871029. A part of the work was done on the Aix Marseille University antivirals platform "AD2P".

## Author contributions

Conceptualization, J.S.D., M.C., G.M., and A.N.; methodology, J.S.D., M.C., G.L., G.M., C.L., and A.N.; formal analysis, J.S.D., M.C., and G.L.; investigation, J.S.D., M.C., G.M., F.T., P.R.P., G.P., K.B., C.L., and A.N.; resources, F.T., B.C., J.G., X.d.L., C.S., and A.N.; writing—original draft, J.S.D., M.C., C.L., J.G., C.S., and A.N.; writing—review & editing, J.G., X.d.L., C.S., and A.N.; visualization, J.S.D., M.C., G.L., F.T., P.R.P., and A.N.; supervision, A.N.; funding acquisition, F.T., B.C., X.d.L., and A.N.

## Competing interests

J.G. has consulted for F. Hoffman-La Roche. C.S. has consulted for ViiV Healthcare, MSD and Gilead. The remaining authors declare no competing interests.
