## [Peer Review File · Nature Communications]

REVIEWER COMMENTS

Reviewer #1 (Remarks to the Author):

Favipiravir antiviral efficacy against SARS-CoV-2 in a hamster model

NCOMMS-20-28616

Driouich et al. evaluated the prophylactic and therapeutic efficacy of Favipiravir in the hamster model of SARS-CoV-2. They discovered that the combination of high doses of favipiravir with low doses of virus were effective in reducing lung virus titers and weight loss. Larger amount of virus or lower doses of the favipiravir showed minimal or no effect of said compound, which is in line with a report on BioRxiv. The authors also quantified the amounts of Favipiravir in plasma and the lungs of treated animals and observed high concentrations ($>$ in vitro EC50 value) of the drug 1-5 hours post infection in the plasma and lungs. Finally, they compared the genome sequence between CoV-2 isolated from mock-treated or treated animals and found an increase in the number of non-synonymous mutations and a decrease in the frequency of synonymous mutations in the Favipiravir treated animals.

Overall, the study is well done and the manuscript is well written. Below are a list of comments and suggestions that will need to be addressed.

- Line 15: Words like “dramatic” should be avoided in the abstract, especially when this applies to a toxic dose (75 mg/day) of Favipiravir or only with a low challenge dose.
- Reporting molarities instead of grams for favipiravir would make it easier to compare with other publications.
- The normalization of the weight loss is unusual and the data is mostly driven by the weight gain of the mock infected animals and not by the weight loss of the infected ones (for example see below). Also the description in the legends is missing $\times 100$. Typically a hamster will gain ~ 15 - 20% weight in six days at that age. A real weight loss of 10% would be normalized to $90\%/120\% = 0.75$, while 5% real weight loss would be 79% normalized. Therefore this normalization leads to an overestimation of the disease and weight loss in these animals and the actual weight loss (% of initial weight at each day) should be presented.
- Figure 2b occasionally only has 5 animals in the high dose group. What happened to the 6th one.
- Include the type of statistical method that was used for the analysis of all the data.
- Figure 3: include the number of animals per condition in the legends. Also do not use the normalized values per explanation above.

- The lung concentrations are similar to those of the plasma. Where the animals perfused prior to organ collection and testing of drug concentration? If not, the authors should acknowledge this in the results that the lung concentration could be reflective of the residual plasma.
- The description of the PK data is very detailed and not easy to understand. The non-linear relationship also seems to be dependent on the time of sampling since at 30 min, the concentration appears directly correlated with the dose of administration.
- What are the numbers in Fig 5a? Why were only 4 animals tested and how were these selected from the 6 animals in Fig 2b?
- Explain why Favipiravir only increases non-synonymous mutations and not synonymous. Is this because A and G is more frequent at the 1st and 2nd position of the codon? It is hard to understand why the drug only increases non-synonymous mutations instead of random nucleotides.
- In the sequence analysis, have you corrected for the lower input of viral RNA in the 37 and 75 mg dose? What is the impact of more PCR cycles on the mutation frequency?
- Line 221 mentions frequency of 10%, but Fig 5a and b do not show this.
- Line 279: The high challenge dose would be the other explanation for the modest effect in reference 21 (Fig 2).
- Line 287: the C-U mutations were not significantly different. Is the bias towards G-A previously reported and what could cause that?
- Line 320-326: please reference the figures that these statements are based on, i.e. dose, therapeutic versus prophylactic.
- Line 420: what is the target of you primers and probes for qPCR?

Finally, the data presented in supplementary figure 1 should be in the main manuscript.

Reviewer #2 (Remarks to the Author):

The authors describe the characteristics of their SARS-CoV2 infection model in hamsters and assess the effect of favipiravir in this model. They demonstrate that favipiravir, at high doses, that result also in (some) toxicity, results in an important reduction of infectious virus titers in the lungs and an improvement of the clinical condition. The pharmacokinetics were determined and it is shown that an exposure can be reached that is comparable to the exposure in humans that were treated with high doses of favipiravir in clinical studies of the drug against Ebola. Finally, the authors also

demonstrate – in line with the known mechanism of action of the drug against flu and some other viruses- that the antiviral activity can be linked with a increase in the mutation rate of the virus and as a (likely) consequence a decrease of the relative infectivity. The finding that the drug can markedly reduce viral levels in the lungs (albeit at high doses) is relevant and of importance. The study has however several shortcomings.

Major comments

The rather rudimentary implementation of the hamster model does not bring any extra to other studies in which infection of hamsters with SARS-CoV2 has been described. Hence these data should be moved to the supplemental section. Also it is not indicated in the legend how many animals were used per condition/datapoint.

Fig 2 How was the drug tolerated with this dosing scheme? It is mentioned in the text that n=6 hamsters/condition but this should also be in the Legend. I believe that it would be better to combine “Lung infectious titers”; “Lung viral RNA yields” and “Infectivity ratio” in one and the same panel (3 bars/condition). I assume that these data are from one single experiment, which is far from ideal. This should at least explicitly be mentioned. In the Legend the calculation of “relative lung viral particle infectivity” is explained, (also in Fig4); it is sufficient to have this in the M&M.

It is disturbing that there is no consistency in the way the effect of the drug is assessed in the experiments presented in Fig 2 and 3. Why was there no clinical follow-up of the animals in the exp presented in Fig2? It is also disappointing that the effect of the treatment on disease (in particular the condition of the lungs) was not assessed histologically. It would be important to know whether the reduction in infectious virus titers also results in protection of the lungs. Now only the relative body weights are presented. Fig 3: how many animals? Single experiment? BDW s a very rough measure.

Fig 4 same comments as for Fig2; please combine. Why are viral RNA levels now not presented? How many animals?

The entire section on the pharmacokinetics is very descriptive and lengthy. Also the Table 3 in which the PK are reported could move to supplemental section and the key findings of the PK can be briefly described in the results section. As such, a PK measurement does not really provide exciting science. In lines 179-182: the phrasing/formulation is unclear. Finally and importantly, the animals were not perfused before the lungs were isolated to quantify drugs levels in the tissue (a piece of tissue was rinsed in PBS); this does not allow to accurately determine drug concentration. Therefore a disclaimer should be added.

Supplemental Fig 1: There are no legends to the graphs. Middle graphs 'y-axis' is labeled "% inhibition" ... of what...? How many experiments (seems like one; which is not acceptable). Same comments for the lower panel;

Supplemental Fig2; nog legend.

Supplemental Fig 3 and 4: number of animals?

Minor comments

Besides EC50 and CC50 best to present also EC99 and CC99

In the results section line 91 the definition of the "pre-emptive" condition is given. This is however not clearly done for the "preventive" condition. It looks as if this should have been done on line 126. All by all, I find the terminology not well chosen and there is technically also not so much difference between the two conditions. A direct comparison of the efficacy of the two conditions (see also comments above) is not possible, given the different way the experiments have been analyzed.

I don't think that the wording "plasmatic viral loads" is correct. Best is to replace "plasmatic" by "plasma".

Legend Fig 4 correct "awee"

Reviewer #3 (Remarks to the Author):

Driouich and coworkers investigate the efficacy of favipiravir in treatment of SARS-CoV-2 infection in Syrian hamsters. The drug was given either before, at the time of, or after infection. In addition, pharmacokinetic data in the hamster are provided that show reasonably good biodistribution in this widely used rodent model. The results of the study are that favipiravir shows limited antiviral activity in vitro and in vivo at high concentrations of approximately 450 μ M, less so at ~250 μ M. In addition, it is reported that "prophylactic" treatment is superior to metaphylactic and therapeutic application. Importantly, antiviral activity is correlated with an increase in virus mutation rates, again most

pronounced at high concentrations of the drug, which is largely driven by G/A and less so by C/U mutations.

The preclinical evaluation of drugs repurposed for SARS-CoV-2 is an important endeavour in the global fight against COVID-19; unfortunately, favipiravir seems not to be a candidate that holds a lot of promise based on the data presented. In addition, some of the experiments conducted and the results raise some substantial questions and concerns.

Major:

1) While different EC50 for favipiravir are reported in the literature, the EC50 reported here for the drug for SARS-CoV-2 in Vero E6 cells was greater than 500 μ M - if I interpret lines 56-62 correctly. It should be explained why and how the authors decided to move to in vivo experiments in the light of the in vitro findings.

2) Figure 1 shows the establishment of the Syrian hamster infection model. However, there are numerous reports of this model reported in the literature. The authors report about 10^4 to 10^6 "viral copies" in the lung and peak copy numbers of $>10^7$ in the plasma. First, one would expect a point of reference (10^4 to 10^6 and 10^7 , respectively) for these measurements. Was this per mg/or μ L? Also, "viral copies" is a misnomer, it should be virus genome copies and "plasmatic" also should be changed

3) lines 87 and throughout the manuscript: it would be easier for the reader if drug concentrations were given in μ M and not mg.

4) Most concerning are Figure 2 - 4 for a number of reasons:

a) There is only (limited) efficacy of favipiravir in doses that seem to be toxic for the animals. How is it possible to perform experiments with a substance that shows toxicity. How did you determine toxicity and how did that impact for example on medium body weights. It is absolutely essential to show the body weight developments relative to the mock animals and not just within the group (Figure 3)!

b) How do you explain that RNA copy numbers are LOWER than the virus copy numbers? That is very unusual. In this context, particle/pfu ratios of <1 are theoretically impossible

c) Again, the fact that virus loads in plasma exceed those in the lungs is remarkable and inconsistent with published data. This discrepancy between this and many other studies needs to be discussed.

5) Lines 327 and on: The discussion of the dosage determined here to be effective (which, I might add, is toxic) to be in the "same range" as effective dosages in humans against Ebola is misleading. By the authors on account, doses of 1400 mg/kg/day were applied in the hamster, while 1200 mg/day were applied to humans (6000 mg on day one). Hence, we are talking about a roughly 100-fold higher dose used in the hamster

**The reviewers,
*Nature Communications***

Marseille, November 09th 2020

First of all, we would like to express our gratitude to the reviewers, who provided a thorough review of our work. The comments raised have helped us to improve the manuscript significantly.

The following represents a reply to each of the comments and then refers to the changes made in the revised manuscript.

Major modifications of the manuscript

Pathology analysis:

The lack of histological analysis was a weak point in our work. Additional experiments were conducted to provide answers on the histology of the lungs with and without treatment. Results of these experiments were added to the manuscript. Briefly, groups of 4 hamsters were infected with 10^4 TCID₅₀ of virus and treated with 37.5 or 75mg/day of favipiravir following a preventive or a preemptive strategy (in comparison with a group of untreated hamsters). Entire lungs were harvested at day 3 and 5 post-infection (dpi). Different anatomic pulmonary compartments were evaluated and a score was assigned based on severity. We found that lungs of hamsters treated with both doses displayed significantly decreased lesions when compared to the corresponding untreated group. These results are totally in accordance with our previous results presented in the original manuscript and further bring another argument in favour of the efficacy of high doses of favipiravir against Sars-Cov2 in a hamster model.

Unclear significance of the viral titer data:

The concerns raised by the reviewers regarding viral titers data came mostly from confusion around originals Figure 2 and 4 (now Figure 1 and 3). Therefore, we agree that the first version of these figures was unclear and many modifications have been made. First, a thin band of white has been added around each icons to ensure their visibility. We also added the unity of each of the three entities ("Lung infectious titers", "Lung viral RNA yields" and "Plasma viral loads") in the title of the y-axis. In original Figure 2, the x-axis was divided into three parts in order to enhance the independence of the experiments according to each dose of virus used for infection. In order to increase the clarity of the figures, each panel (each "line") were identified individually with the letters: "b", "c", "d", "e" and "f" (with corresponding legends). Finally, we also added a panel representing the clinical course of the disease (based on animal's weights, expressed as the % of initial weight at each day) for each of these experiments according to

the dose of virus used for infection. We hope these modifications will help the reviewers and the future readers to better understand these data. Especially, we hope that result section and these figures now clearly show (i) that data presented come from independent experiments, (ii) that lung viral RNA yields (determined using a RT-qPCR assay) are higher than lung infectious titers (determined using a TCID₅₀ assay), (iii) that virus loads in plasma do not exceed those in the lungs and (iv) that overall, all these modifications bring more consistency in the way the effect of the drug is assessed.

Toxic dosage in hamsters:

The toxicity observed in our study with the highest dose of favipiravir is obviously a concern. Consequently, we modified the abstract and clearly presented the signs of toxicity in the preemptive and the preventive strategies experiments.

However, clear efficacy of the molecule was demonstrated using the medium dose (37.5mg/day) which induced very little signs of toxicity (only when uninfected animals were treated more than three days): we observed significant reduction of infectious titers in lung using the preemptive strategy (with doses of 10⁵ and 10⁴ TCID₅₀ of virus) and using the preventive strategy. We also demonstrated that this dose allowed clinical alleviation of the disease after infection of hamster with 10⁵ and 10⁴ TCID₅₀ of virus. Results of histological analysis also corroborated these findings (see above) and reinforced the robustness of our conclusions.

Using a toxic dose to demonstrated efficacy of a treatment may seem inappropriate. However, our objective was not to assess the efficacy of favipiravir only using this high dose alone (75mg/day). We aimed to determine *in vivo* dose-response curves in order to estimate drug 50%, 90% and 99% effective doses. Therefore, we used three different doses of favipiravir, including a high dose associated with sign of toxicity. In addition, determination of pharmacokinetics parameters was also done using these three doses in order to make assumptions on the capacity of the molecule to be effective in SARS-COV-2 infected patients.

Moreover, a recent study (*Favipiravir at high doses has potent antiviral activity in SARS-CoV-2-infected hamsters, whereas hydroxychloroquine lacks activity. Kaptein et al. 2020*) showed that favipiravir at 1,000 mg/kg/day BID could have a beneficial impact on the disease (including lung viral RNA yields, lung infectious titers and histological lung pathology) with no toxicity signs observed in hamsters of 6 to 10 weeks old.

Overall we think that subsequent studies are essential to determine if a tolerable dosing regimen could generate similar exposure in non-human primates, associated with significant antiviral activity, before testing a high dose regimen in COVID-19 patients.

Reviewer #1 (Remarks to the Author):

Drriouch et al. evaluated the prophylactic and therapeutic efficacy of Favipiravir in the hamster model of SARS-CoV-2. They discovered that the combination of high doses of favipiravir with low doses of virus were effective in reducing lung virus titers and weight loss. Larger amount of virus or lower doses of the favipiravir showed minimal or no effect of said compound, which is in line with a report on BioRxiv. The authors also quantified the amounts of Favipiravir in plasma and the lungs of treated animals and observed high concentrations (> *in vitro* EC50 value) of the drug 1-5 hours post infection in the plasma and lungs. Finally, they compared the genome sequence between CoV-2 isolated from mock-treated or treated animals and found an increase in the number of non-synonymous mutations and a decrease in the frequency of synonymous mutations in the Favipiravir treated animals.

Overall, the study is well done and the manuscript is well written. Below are a list of comments and suggestions that will need to be addressed.

We thank the reviewer for the positive comments on the paper.

• **Line 15: Words like “dramatic” should be avoided in the abstract, especially when this applies to a toxic dose (75 mg/day) of Favipiravir or only with a low challenge dose.**

We agree with the reviewer. The word « dramatic » has been removed from the abstract (line 16)

• **Reporting molarities instead of grams for favipiravir would make it easier to compare with other publications.**

We thank the reviewer for pointing this out. We amended the *in vitro* evaluation of favipiravir efficacy section and replaced concentration by corresponding molarities (lines 60, 62, 63, 64, 65 and 66).

• **The normalization of the weight loss is unusual and the data is mostly driven by the weight gain of the mock infected animals and not by the weight loss of the infected ones (for example see below). Also the description in the legends is missing x100. Typically a hamster will gain ~15-20% weight in six days at that age. A real weight loss of 10% would be normalized to $90\%/120\% = 0.75$, while 5% real weight loss would be 79% normalized. Therefore this normalization leads to an overestimation of the disease and weight loss in these animals and the actual weight loss (% of initial weight at each day) should be presented.**

We thank the reviewer for pointing the complexity of our calculation for normalized weights. Therefore we have changed all figures representing animal weights. As suggested by the reviewer, we have now presented the % of initial weight at each day and specified it in the legends of all figures. Moreover, we removed the expression « weight loss » in the manuscript as it was no longer appropriate.

• **Figure 2b occasionally only has 5 animals in the high dose group. What happened to the 6th one.**

The original figure 2 (now figure 1) has given rise to numerous comments, therefore many modifications have been made (see paragraph “Unclear significance of the viral titer data” in the “Major modifications of the manuscript” section). Concerning the number of icons in the figure, six of them, representing each of the 6 animals, were represented in all cases in the figure. However, some of them were not perfectly visible, in particular hidden by the width of the contours or by the x-axis. In all figures, a thin band of white has been added around each icons to ensure their visibility.

• **Include the type of statistical method that was used for the analysis of all the data.**

All detailed statistical methods and their results are presented in the supplemental data : Statistical analysis for the implementation of the hamster model are presented in supplemental table 1 ; Statistical analysis of other *in vivo* experiments are presented in supplemental table 3, 4, 5 and 6 ; Statistical analysis of the histopathological experiments are presented in supplemental table 8. Statistical analysis of the mutagenic effect of favipiravir are presented in supplemental table 11 and 12.

• **Figure 3: include the number of animals per condition in the legends. Also do not use the normalized values per explanation above.**

We thank the reviewer for pointing this lack. The number of animals per condition in each experiments was added in the legends of all figures. All figures representing animal weights were also modified as explain above.

• **The lung concentrations are similar to those of the plasma. Where the animals perfused prior**

to organ collection and testing of drug concentration? If not, the authors should acknowledge this in the results that the lung concentration could be reflective of the residual plasma.

We thank the reviewer for this comment. Organ collection was done at different time points according to the PK study as described in the results section. In the non-infected animals, favipiravir was measured at 0.5 h or 5 hours after I.P administration and for infected animals at 12-hours after administration. Therefore, lung concentrations reflect the drug concentration at each specific time point and indeed this reflects the residual concentration for the infected animals only. This precision was added in the results section (Line 234-235): "Favipiravir trough concentration was quantified in plasma (n=9) and lung tissue (n=3)" and in the legend of Table 3.

• The description of the PK data is very detailed and not easy to understand. The non-linear relationship also seems to be dependent on the time of sampling since at 30 min, the concentration appears directly correlated with the dose of administration.

We thank the reviewer for this relevant comment. The description of the PK data have been well detailed in order to understand two major concerns:

- first, to exhibit the non-linearity of the favipiravir PK, mainly due to the inhibition of its own metabolism
- second, to evaluate the tissue distribution as it is a major issue to evaluate the efficacy of the drug against SARS-coV-2 virus infection.

Taken together, these 2 points are crucial to determine the effective dosing regimens that could be further evaluated in both NHP models and humans.

Indeed, as mentioned in your comment, the concentration at 30 min appears to be correlated to the dose probably because it is measured after only a single administration at the time of the maximum concentration. However, this correlation is no longer observed at the other time points. We determined the PK after single and multiple doses to well demonstrate this non linearity and understand the complexity of the drug PK. This complexity is a key issue to optimize the choice of the dosing regimen but also the protocol of administration.

As observed in this study, the best results were obtained in a preventive strategy with the two highest doses regimens and this is may be mainly explained by the drug PK.

• What are the numbers in Fig 5a? Why were only 4 animals tested and how were these selected from the 6 animals in Fig 2b?

The numbers in Figure 5a represent the animal IDs. In the revised version of the manuscript, we have added the sign « # » and the legend « Animal ID » in the figure to clarify it.

The mutagenic analysis was made with four animals per group based on our personal experience in assessing this kind of effect with other RNA viruses. We selected the first animals for each group based on animal identity number.

• Explain why Favipiravir only increases non-synonymous mutations and not synonymous. Is this because A and G is more frequent at the 1st and 2nd position of the codon? It is hard to understand why the drug only increases non-synonymous mutations instead of random nucleotides.

Favipiravir induced appearance of large number of mutations (mostly G→A and C→U) randomly distributed into viral genomes. These mutations were equally located in the 1st, 2nd and 3rd positions of codons. Consequently, mutations at 1st and 2nd codon position (mostly non-synonymous based on genetic code) represent around 2/3 of all mutations while those at 3rd position (mostly synonymous based on genetic code) represent 1/3 of all mutations. Thereby, this mutagenic effect mechanically

increase the proportion of non-synonymous mutations. This observation suggests that no compensatory mechanism was triggered by the virus to eliminate them (*i.e.* negative selection).

• In the sequence analysis, have you corrected for the lower input of viral RNA in the 37 and 75 mg dose? What is the impact of more PCR cycles on the mutation frequency?

We thank the reviewer for this comment. However, we have not corrected for the lower input of viral RNA in the 37 and 75mg dose. Indeed, animal were infected with 10^6 TCID₅₀ and at this dose, viral RNA yields differences were lower than 1 log₁₀ between all the samples sequenced (original Figure 2c, now Figure 1c). In addition, the impact on mutation frequency is very low when highly positive samples as clarified lung homogenates are sequenced (ct values of around 20).

• Line 221 mentions frequency of 10%, but Fig 5a and b do not show this.

We thank the reviewer for highlighting the lack of coherence between the manuscript and the figure. Figure 5a shows that almost all mutations exhibited a frequency lower than a frequency of 0.1 (which is 10%): each triangle represents a mutation and a majority of them were below the line of 0.1. We have modified the y-axis numbering in order to increase the clarity of this figure. Figure 5b shows that mutations were distributed throughout the whole genome. We have also amended the sentence (lines 272-275) to be more explicit: « Overall, no majority mutations were detected (mutation frequency >50%) and almost all of the mutations exhibited a frequency lower than 10% mutations (Figure 5a). In addition, mutations were distributed throughout the whole genome (Figure 5b). »

• Line 279: The high challenge dose would be the other explanation for the modest effect in reference 21 (Fig 2).

We agree with this comment. This paragraph (lines 334-348) has been modified since the publication of this other study.

• Line 287: the C-U mutations were not significantly different. Is the bias towards G-A previously reported and what could cause that?

We agree with the reviewer. In the revised manuscript, we have deleted these words (line 356).

The bias towards G-A was previously reported in a study establishing the antiviral efficacy of favipiravir against ebola virus (*Antiviral efficacy of favipiravir against Ebola virus: A translational study in cynomolgus macaques. Guedj et al. 2018*). It could be explain by the mecanism of action of the molecule. The active metabolite of favipiravir acting as a purine nucleotide (this is now indicated in the manuscript, lines 349-350) may induce G→A mutations during positive-strand RNA synthesis, whereas C→U mutations are predicted to result from incorporation of the metabolite during negative-strand RNA synthesis (*T-705 (Favipiravir) Induces Lethal Mutagenesis in Influenza A H1N1 Viruses In Vitro. Baranovich et al. 2013*).

• Line 320-326: please reference the figures that these statements are based on, i.e. dose, therapeutic versus prophylactic.

We thank the reviewer for this comment. In the revised manuscript references to tables and figures were added in the discussion section. We also added more details in order to increase the clarity of these statements (ie, doses of virus, therapeutic vs prophylactic) (lines 389-398).

• Line 420: what is the target of you primers and probes for qPCR?

The gene targeted by our primers/probes RT-qPCR system is the RdRp coding regions as described in supplemental table 9.

Finally, the data presented in supplementary figure 1 should be in the main manuscript.

We thank the reviewer for pointing this out. As mentioned below (see section “List of all modifications of the manuscript” line 86), the table from the original supplementary table 1 was added in the main manuscript.

Reviewer #2 (Remarks to the Author):

The authors describe the characteristics of their SARS-CoV2 infection model in hamsters and assess the effect of favipiravir in this model. They demonstrate that favipiravir, at high doses, that result also in (some) toxicity, results in an important reduction of infectious virus titers in the lungs and an improvement of the clinical condition. The pharmacokinetics were determined and it is shown that an exposure can be reached that is comparable to the exposure in humans that were treated with high doses of favipiravir in clinical studies of the drug against Ebola. Finally, the authors also demonstrate – in line with the known mechanism of action of the drug against flu and some other viruses- that the antiviral activity can be linked with a increase in the mutation rate of the virus and as a (likely) consequence a decrease of the relative infectivity. The finding that the drug can markedly reduce viral levels in the lungs (albeit at high doses) is relevant and of importance. The study has however several shortcomings.

Major comments

The rather rudimentary implementation of the hamster model does not bring any extra to other studies in which infection of hamsters with SARS-CoV2 has been described. Hence these data should be moved to the supplemental section. Also it is not indicated in the legend how many animals were used per condition/datapoint.

We thank the reviewer for pointing this out. As mentioned below (see section “List of all modifications of the manuscript” line 86), the original Figure 1 on the implementation of the hamster model was moved to the supplementary section. We thank the reviewer for pointing this lack. The number of animals per condition in each experiments was added in the legends of all figures.

Fig 2 How was the drug tolerated with this dosing scheme? It is mentioned in the text that n=6 hamsters/condition but this should also be in the Legend. I believe that it would be better to combine “Lung infectious titers”; “Lung viral RNA yields” and “Infectivity ratio” in one and the same panel (3 bars/condition). I assume that these data are from one single experiment, which is far from ideal. This should at least explicitly be mentioned. In the Legend the calculation of “relative lung viral particle infectivity” is explained, (also in Fig4); it is sufficient to have this in the M&M.

We thank the reviewer for these comments. The original Figure 2 (now Figure 1) has given rise to numerous comments, therefore many modifications have been made (see paragraph “Unclear significance of the viral titer data” in the “Major modifications of the manuscript” section). Concerning the evaluation of drug tolerance with this dosing scheme, we have added figures representing animal weights for all these experiments. Overall, only animals treated with the highest dose of favipiravir showed signs of toxicity (specified lines 114-116). The number of animals per condition (n=6) was added to the legend.

If we understand well the reviewer’s proposal regarding the combination of “Lung infectious titers”, “Lung viral RNA yields” and “Infectivity ratio” in one and the same panel, it seems to us unfeasible to gather these three entities of different natures: « TCID₅₀/copy of γ -actine gene », « viral genome copies/copy of γ -actine gene » and « TCID₅₀/ viral genome copies ».

We thank the reviewer for pointing the lack of precision concerning repeatability (and reproducibility) of our work. We have amended the manuscript to be more explicit (lines 93-94). As mentioned in the revised version of the manuscript, data presented in original Figure 2 come from three independent experiments (with 10^6 , 10^5 and 10^4 TCID₅₀ of virus during infection). Overall, we found consistency between our results, as the efficacy of the treatment increased (i) with the dose of favipiravir and (ii) when decreasing the dose of virus. The efficacy of the preventive antiviral therapy was also assessed in an independent experiment and gave results coherent with those obtained using a preemptive strategy. Altogether, we believe that all these results brought enough arguments to avoid duplicating all experiments, especially when respecting the three Rs (Replacement, Reduction, Refinement) principles for animal experimentation. In addition, our results were in accordance with another study conducted by Kaptein, S. J. et al. published in PNAS (*Favipiravir at high doses has potent antiviral activity in SARS-CoV-2-infected hamsters, whereas hydroxychloroquine lacks activity. 2020.*).

We disagree with the reviewer concerning the positioning of the explanation of the calculation of “relative lung viral particle infectivity”. Since this article could be published in *Nature Communications* described as an open access journal that publishes high-quality research from all areas of the natural sciences, and not intended for virologists only, we believe that this definition will be useful for many readers.

It is disturbing that there is no consistency in the way the effect of the drug is assessed in the experiments presented in Fig 2 and 3. Why was there no clinical follow-up of the animals in the exp presented in Fig2? It is also disappointing that the effect of the treatment on disease (in particular the condition of the lungs) was not assessed histologically. It would be important to know whether the reduction in infectious virus titers also results in protection of the lungs. Now only the relative body weights are presented. Fig 3: how many animals? Single experiment? BDW s a very rough measure.

The original Figure 2 (now Figure 1) has given rise to numerous comments, therefore many modifications have been made (see paragraph “Unclear significance of the viral titer data” in the “Major modifications of the manuscript” section). Concerning the clinical follow up of the animals, we have added new panels in originals Figures 2 and 4 (now Figures 1 and 3) representing mean animal weights during the course of the experiments. All individual weights of animal are also presented in supplemental table 2. Because, the same experimenters carried out infection/treatment/clinical follow-up, it was impossible to perform a blind trial. Therefore it appeared to us that the observation of mild symptoms could have been biased. This is why we have presented only the weight of the animals.

We agree with the reviewer. As mentioned above (see paragraph “Pathology analysis” in the “Major modifications of the manuscript” section), the lack of histological analysis was a weak point in our work. Additional experiments were conducted to provide answers on the histology of the lungs with and without treatment. These works were added to the manuscript (lines 183-205).

We also thank the reviewer for pointing the roughness surrounding the relative body weights measurement. We have changed all figures representing animal weights, they now represent the % of initial weight at each day.

The number of animals per condition in original Figure 3 (now Figure 2) was added in the legend.

As mentioned in the revised version of the manuscript (line 142), data presented in original Figure 3c and 3d come from two independent experiments (with 10^5 and 10^4 TCID₅₀ of virus during infection). Overall, we found consistency between our results. We believe that all these results brought enough arguments to avoid duplicating all experiments, especially when respecting the three Rs (Replacement, Reduction, Refinement) principles for animal experimentation.

Fig 4 same comments as for Fig2; please combine. Why are viral RNA levels now not presented? How many animals?

We thank the reviewer for these comments. Concerning the first remark, see answers above.

We disagree with the reviewer, the viral RNA levels are represented on original Figure 4 (now Figure 3). However, this remark is probably the sign of a lack of clarity regarding the presentation of this figure. Consequently, for more consistency and clarity, we have modified the original Figure 4 in the same manner that we have modified the original Figure 2 (now Figure 1).

We thank the reviewer for pointing this lack. The number of animals per condition in original Figure 4 was added in the legend.

The entire section on the pharmacokinetics is very descriptive and lengthy. Also the Table 3 in which the PK are reported could move to supplemental section and the key findings of the PK can be briefly described in the results section. As such, a PK measurement does not really provide exciting science. In lines 179-182: the phrasing/formulation is unclear. Finally and importantly, the animals were not perfused before the lungs were isolated to quantify drugs levels in the tissue (a piece of tissue was rinsed in PBS); this does not allow to accurately determine long concentration. Therefore a disclaimer should be added.

We thank the reviewer for this comment. Some answers have been proposed to help to the comprehension of the PK study, as requested by reviewer 1.

However, we disagree with the reviewer regarding his unfavorable opinion on the PK study. Indeed, in the context of the COVID-19 pandemic, several drugs have been repurposed as potential candidates for the treatment of COVID-19 infection. While preliminary choices were essentially based on *in vitro* potency, clinical translation for most of the drug candidates into effective therapies have been well disappointing. As demonstrated *a posteriori* for some of them, in humans and/or in non-human primates, this may be due to unfavorable *in vivo* pharmacokinetic properties at the doses chosen. Implementation of PK and PD studies is therefore needed in order to increase our comprehension of the dose-exposure-effect relationships of the repurposed drugs. Without these data, efficient evaluation and development of effective dosing regimens will remain difficult. Other scientific societies (Baker et al., 2020) and organizations (Hartman et al., 2020; Rayner et al., 2020; Venisse et al., 2020) have also recently issued call to action for the appropriate application of clinical pharmacology principles in the search for COVID-19 treatments.

That's why, it is particularly relevant to systematically include an appropriate PK study to *in vivo* evaluation in animal models. This provide important data regarding the tissue drug penetration and the time to reach effective drug concentration, both are of major concerns to first understand the dose-response relationship and then determine thoroughly effective dosing regimens.

Regarding the perfusion and organ collection, see response to reviewer 1. Modifications have been done within the manuscript.

Moreover, we think there is a misunderstanding regarding the interpretation of lung concentrations and the objectives. The aim of the study is not to determine the coefficient of partition of the drug but the "real" exposure, particularly at the end of the dosing interval. The drug is not administered continuously, therefore PK in tissue follows generally plasma PK with a lower variability but the exposure is not stable all over the dosing interval. Thus, the lung concentrations reported here accurately reflects the "real" exposure in the animals. Tissues were rinsed using PBS as commonly used by in other PK studies, in order to avoid the blood contamination. Lungs were immediately crushed after collection/rinse and clarified lung homogenates were kept at -80°C until analysis to stop transporters activity.

Supplemental Fig 1: There are no legends to the graphs. Middle graphs 'y-axis' is labeled "% inhibition" ... of what...? How many experiments (seems like one; which is not acceptable).

Same comments for the lower panel;

Supplemental Fig2; nog legend.

Supplemental Fig 3 and 4: number of animals?

We thank the reviewer for pointing this out. Some information in the supplemental Fig 1 has been moved to the main manuscript and legends were added in all supplemental figures. The label of the 'y-axis' « % of inhibition » has been replaced by « viral inhibition (%) ».

For the *in vitro* analysis of favipiravir efficacy, two authors of our study (FT and BC) recently described in *Nature Communication* similar results (*Rapid incorporation of Favipiravir by the fast and permissive viral RNA polymerase complex results in SARS-CoV-2 lethal mutagenesis. 2020. Shannon et al.*). That is why we repeated only once these experiments.

We thank the reviewer for pointing this lack. The number of animals per condition in supplemental Figures 3 and 4 has been added in the legend.

Minor comments

Besides EC50 and CC50 best to present also EC99 and CC99

We disagree with the reviewer. Almost all *in vitro* studies of antiviral efficacy present EC₅₀/EC₉₀/CC₅₀ and do not present EC₉₉/CC₉₉. Moreover, using VeroE6 cells and an MOI of 0.01, it was impossible to calculate EC₉₉ (see supplemental fig 1).

In the results section line 91 the definition of the “pre-emptive” condition is given. This is however not clearly done for the “preventive” condition. It looks as if this should have been done on line 126. All by all, I find the terminology not well chosen and there is technically also not so much difference between the two conditions. A direct comparison of the efficacy of the two conditions (see also comments above) is not possible, given the different way the experiments have been analyzed.

We agree with the reviewer for this comment. The « preventive » condition was assessed in the third set of experiments and its definition is given at the beginning of this section (line 157). We have amended the manuscript in order to indicate that the second set of experiments was carry out using the « preemptive » strategy (line 130).

We have now presented the original Figure 4 (now Figure 3) in the same manner than the original Figure 2 (now Figure 1) in order to facilitate the comparison between the two therapeutic strategies. However, we think that presenting the two strategies in separate figures remains the best way to present our data. Table 2 allows reader to compare overall efficacy between the two conditions.

I don't think that the wording “plasmatic viral loads” is correct. Best is to replace “plasmatic” by “plasma”.

We thank the reviewer for pointing this out. We have replaced the word plasmatic by plasma in the manuscript.

Legend Fig 4 correct “awee”

We thank the reviewer for pointing this out. We have waived this error of syntaxe (line 175).

Reviewer #3 (Remarks to the Author):

Dr. Riouich and coworkers investigate the efficacy of favipiravir in treatment of SARS-CoV-2 infection in Syrian hamsters. The drug was given either before, at the time of, or after infection. In addition, pharmacokinetic data in the hamster are provided that show reasonably good biodistribution in this widely used rodent model. The results of the study are that favipiravir shows limited antiviral activity *in vitro* and *in vivo* at high concentrations of approximately 450µM, less so at ~250µM. In addition, it is reported that "prophylactic" treatment is superior to

metaphylactic and therapeutic application. Importantly, antiviral activity is correlated with an increase in virus mutation rates, again most pronounced at high concentrations of the drug, which is largely driven by G/A and less so by C/U mutations.

The preclinical evaluation of drugs repurposed for SARS-CoV-2 is an important endeavour in the global fight against COVID-19; unfortunately, favipiravir seems not to be a candidate that holds a lot of promise based on the data presented. In addition, some of the experiments conducted and the results raise some substantial questions and concerns.

Major:

1) While different EC₅₀ for favipiravir are reported in the literature, the EC₅₀ reported here for the drug for SARS-CoV-2 in Vero E6 cells was greater than 500 μ M - if I interpret lines 56-62 correctly. It should be explained why and how the authors decided to move to *in vivo* experiments in the light of the *in vitro* findings.

The EC₅₀ in VeroE6 cells reported here were 204 and 446 μ g/mL (at two different MOI). In another study, EC₅₀ was evaluated at 118 μ M using another method (measure of the cytopathic effect) (*Rapid incorporation of Favipiravir by the fast and permissive viral RNA polymerase complex results in SARS-CoV-2 lethal mutagenesis. Shannon et al. 2020*). We moved to *in vivo* experiments because (i) many clinical trials are ongoing, (ii) the mechanism of action of this drug was deciphered with SARS-CoV-2 (Favipiravir exerts an antiviral effect as a nucleotide analogue through a combination of chain termination, slowed RNA synthesis and lethal mutagenesis; *Rapid incorporation of Favipiravir by the fast and permissive viral RNA polymerase complex results in SARS-CoV-2 lethal mutagenesis. Shannon et al. 2020*) and (iii) high doses of this drug were already used in Ebola infected patients.

2) Figure 1 shows the establishment of the Syrian hamster infection model. However, there are numerous reports of this model reported in the literature. The authors report about 10⁴ to 10⁶ "viral copies" in the lung and peak copy numbers of >10⁷ in the plasma. First, one would expect a point of reference (10⁴ to 10⁶ and 10⁷, respectively) for these measurements. Was this per mg/or μ L? Also, "viral copies" is a misnomer, it should be virus genome copies and "plasmatic" also should be changed

We thank the reviewer for pointing this out. The original Figure 1 on the implementation of the hamster model was moved to the supplementary section as mentioned below (see section "List of all modifications of the manuscript" line 86). To increase the clarity of this figure, we modified the legend of the figure: lung viral RNA Yield are expressed in viral genome copies per μ -actine copies and plasma viral loads are expressed in viral genome copies/mL of plasma. The expression "Viral copies" was not used and "plasmatic viral loads" was replaced by "plasma viral loads" as mentioned below.

3) lines 87 and throughout the manuscript: it would be easier for the reader if drug concentrations were given in μ M and not mg.

We thank the reviewer for pointing this out. All concentrations in the *in vitro* evaluation of favipiravir efficacy section were replaced by corresponding molarities as mentioned above (lines 60, 62, 63, 64, 65 and 66).

4) Most concerning are Figure 2 - 4 for a number of reasons:

a) There is only (limited) efficacy of favipiravir in doses that seem to be toxic for the animals. How is it possible to perform experiments with a substance that shows toxicity. How did you determine toxicity and how did that impact for example on medium body weights. It is absolutely essential to show the body weight developments relative to the mock animals and not just within the group (Figure 3)!

The original Figure 2 (now Figure 1) and the problem of toxicity have given rise to important comments, therefore some explanation has been made above (see paragraph "Toxic dosage in hamsters" in the "Major modifications of the manuscript" section). The toxicity was assessed by a clinical follow-up of the animals using weight as the primary criterion. Concerning the evaluation of

drug tolerance with these dosing scheme, we have added panels representing animal weights during these 3 days of treatment.

We also have changed all figures representing animal weights. Weights are now represented in “% of initial weight” at each day and weights of “mock” animals were represented in all figures.

b) How do you explain that RNA copy numbers are LOWER than the virus copy numbers? That is very unusual. In this context, particle/pfu ratios of <1 are theoretically impossible

We disagree with this comment. RNA copy numbers are not lower than virus copy numbers. The originals Figure 2 and 4 (now Figures 1 and 3) show that Lung viral RNA Yields (determined using a RT-qPCR assay) are higher than lung infectious titers (determined using a TCID₅₀ assay). Consequently, the ratios “viral RNA yields/lung infectious titer”, comparable to particle/pfu ratios are always >1. In our figures, we presented the ratios ‘lung infectious titer/ viral RNA yields’ that represented the infectivity of viral particles.

c) Again, the fact that virus loads in plasma exceed those in the lungs is remarkable and inconsistent with published data. This discrepancy between this and many other studies needs to be discussed.

We disagree with this comment. The difference between this two virus loads is due to the fact that loads in plasma are expressed in viral genome copies/mL of plasma and loads in lungs are expressed in viral genome copies per γ -actine copies.

5) Lines 327 and on: The discussion of the dosage determined here to be effective (which, I might add, is toxic) to be in the "same range" as effective dosages in humans against Ebola is misleading. By the authors on accounts, doses of 1400 mg/kg/day were applied in the hamster, while 1200 mg/day were applied to humans (6000 mg on day one). Hence, we are talking about a roughly 100-fold higher dose used in the hamster

We thank the reviewer for pointing this out. We agree with him concerning the dose of favipiravir that is 100-fold higher with hamster in comparison with the one used with human. However, in this paragraph we discussed that at the higher dose used in hamsters (1400mg/kg/day; we agree that toxicity was observed at this dose as showed in results section and as described in discussion section), we observed trough concentrations (*i.e.* the lowest concentration reached by a drug before the next dose is administered) similar to those observed with human using a dose of 1200 mg/day (following a dose of 6000mg at day one).

Because the pharmacokinetics of this molecule is completely different between human and hamster, we think that comparing trough concentrations is more appropriate than comparing dose regimen. However, as mentioned in the discussion section, additional investigations are required to determine whether or not similar favipiravir plasma exposure in SARS-COV-2 infected patients are associated with antiviral activity and with no toxicity.

List of all modifications of the manuscript:

Line 3 and line 7: The new author and his affiliation was added: “Caroline Laprie”.

Line 16: The word “dramatic” was removed.

Lines 18-20: The sentence “The antiviral efficacy observed in this study was achieved with plasma drug exposure comparable with those previously found during human clinical trials and was associated with weight losses in animals” was replaced by “Antiviral efficacy was achieved with plasma drug exposure comparable with those previously found during human clinical trials. Notably, the highest dose of favipiravir tested was associated with signs of toxicity in animals.”

Lines 33-35: The sentence “As of 7 July 2020, more than 11.6 million cases of COVID-19 have resulted in more than 538,000 deaths” was updated: “As of 6 November 2020, more than 48.8 million cases of COVID-19 have resulted in more than 1,235,000 deaths”.

Lines 60, 62, 63, 64, 65 and 66: Concentrations expressed in “ $\mu\text{g}/\text{mL}$ ” are now expressed in “ μM ”.

Line 61: The words “Table 1 and” was added.

Line 67: The table originally presented in supplemental figure 1 (“Summary of 50-90% drug effective concentrations and Infectious titer reductions”) was incorporated in the manuscript.

Line 77: “Figure 1a” was replaced by “Figure S2”.

Line 78: “Figure 1b” was replaced by “Figure S2”.

Line 81: “Figure 1a and 1b” was replaced by “Figure S2 and Table S1”.

Line 85: The expression “significant weight losses” was replaced by “normalized weights (i.e. % of initial weights) significantly lower”.

Line 86: “Figure 1c” was replaced by “Figure S2”.

Line 86: The original figure 1 (“Implementation of hamster model”) was moved to the supplemental section. To increase the clarity of this figure, we modified the legend of the figure: (i) lung viral RNA Yield are expressed in viral genome copies per γ -actine copies, (ii) plasma viral loads are expressed in viral genome copies/mL of plasma, (iii) the expression “plasmatic viral loads” was replaced by “plasma viral loads”, (iv) the number of hamsters used was added and (v) the initial calculation for normalized weights was simplified in “% of initial weight of the animal at day n ”. Instead we have incorporated the table originally presented in supplemental figure 1 (“Summary of 50-90% drug effective concentrations and Infectious titer reductions”) in the manuscript.

Line 93: “Figure 2a” was replaced by “Figure 1a”.

Lines 93-94: The sentence “Each dose of virus was assessed in an independent experiment.” was added.

Line 97: “Figure 2b” was replaced by “Figure 1b”.

Line 106: “Figure 2b” was replaced by “Figure 1c”.

Line 112: “Figure 1d” was added in the sentence.

Line 121: The expression “plasmatic viral loads” was replaced by “plasma viral loads”.

Line 114: “Figure 2b” was replaced by “Figure 1e”.

Lines 114-116: The sentence “Finally, signs of toxicity were observed with animal treated with the dose of 75mg/day TID: normalized weights were significantly lower than those of untreated animals (Figure 1f).” was added.

Lines 117-128: The original figure 2 was modified (see paragraph “Unclear significance of the viral titer data” in the “Major modifications of the manuscript” section). In the legend of the initial figure 2, (i) the number of hamsters used was added, (ii) the expression “plasmatic viral loads” was replaced by “plasma viral loads”, (iii) the legend of the last panel “f Clinical course of the disease. Normalized weight at day n was calculated as follows: % of initial weight of the animal at day n .” was added and (iv) supplemental table related to the figure was updated (Table S2, S3 and S4). Finally, the legend of the figure was organized with legends of panels a, b, c, d, e and f.

Line 129: In the legend of Table 2, “Figure S2” was replaced by “Figure S3”.

Lines 130-131: The word “treatment” was replaced by “the preemptive therapy”.

Line 131: The word “loss” was removed.

Line 131: "Figure 3a" was replaced by "Figure 2a".

Lines 131-132: The word "Beforehand" was replaced by "Since signs of toxicity were noticed during the first set of experiments".

Line 133: The expression "from day 0 to day 3" was replaced by "during four days".

Line 133: "Figure 3b" was replaced by "Figure 2b".

Lines 133-135: The sentence "High toxicity was observed with the dose of 75mg/day TID with significant weight loss noticed from the first day of treatment (Table S4)." was replaced by " High toxicity was observed with the dose of 75mg/day TID with, from the first day of treatment, normalized weights significantly lower than those of untreated animals (Table S5)."

Line 136: The expression "4 and 5" was replaced by "4, 5 and 6".

Line 138: The expression "weight losses" was replaced by "normalized weights".

Line 140: "Figure S3" was replaced by "Figure S4".

Line 142: The sentence "Each dose of virus was assessed in an independent experiment." was added.

Line 144: "Figure 3a" was replaced by "Figure 2a".

Line 145: "Figure 3c-d" was replaced by "Figure 2c-d".

Lines 148-156: To increase the clarity of the original figure 3, we modified the panel as weights are now expressed in "% of initial weight of the animal at day *n*". In the legend of the original figure 3, (i) the number of hamsters used was added, (ii) the initial calculation for normalized weights was simplified in "% of initial weight of the animal at day *n*" and (iii) supplemental table related to the figure was updated (Table S2, S4 and S5).

Line 159: "Figure 4a" was replaced by "Figure 3a".

Line 160: "Figure 4b" was replaced by "Figure 3b".

Line 165: "Figure 3c" was added.

Line 167: "Figure 3e" was added.

Line 167: The expression "plasmatic viral loads" was replaced by "plasma viral loads".

Line 168: "Figure 3f" was added.

Lines 168-170: The sentence "Once again, signs of toxicity were observed with animal treated with the dose of 75mg/day TID: normalized weights were significantly lower than those of untreated animals (Figure 3d)." was added.

Lines 171-182: The original figure 4 was modified in the same manner as the original figure 2 (see paragraph "Unclear significance of the viral titer data" in the "Major modifications of the manuscript" section). In the legend of the initial figure 4, (i) the number of hamsters used was added, (ii) the expression "awee" was removed, (iii) the expression "plasmatic viral loads" was replaced by "plasma viral loads", (iv) the expression ". They are" was replaced by "are", (v) the legend of the last panel "d Clinical course of the disease. Normalized weight at day *n* was calculated as follows: % of initial weight of the animal at day *n*." was added and (vi) supplemental table related to the figure was updated (Table S2, S3 and S4). Finally, the legend of the figure was organized with legends of panels a, b, c, d, e and f.

Lines 183-204: A paragraph on the histological analysis was added: "In a last set of experiments, we assessed the impact of favipiravir treatment on lung pathological changes induced by SARS-CoV-2. Animals were intranasally infected with 10^4 TCID₅₀ of virus. Treatment with two doses of favipiravir (37.5 and 75mg/day TID) was initiated one day before infection (preventive antiviral therapy) or at day of infection (preemptive antiviral therapy) and ended at 3 dpi. For each therapeutic strategy and for

each dose of favipiravir, a group of four animals was sacrificed at 3 and 5 dpi (Figure 4a and 4c). As a control, we used four vehicle-treated groups of four animals (one at 3 dpi and one at 5 dpi for each therapeutic strategy). Based on severity of inflammation, alveolar hemorrhagic necrosis and vessel lesions, a cumulative score from 0 to 10 was calculated and assigned to a grade of severity (0=normal ; 1=mild ; 2=moderate ; 3=marked and 4=severe; details in Table S7). Overall, lungs of untreated animals displayed typical lesions of air-borne infection (i.e. broncho-interstitial pneumonia), with a progression between 3 dpi and 5 dpi that reflects the virus dissemination within the respiratory tree as previously demonstrated. At 3 dpi, 7/8 untreated animals displayed mild pulmonary pathological changes (Figure 4b and 4d) leading to difficulty to assess the efficacy of the treatment even if almost all mean cumulative scores of treated animals were significantly lower than those of untreated groups. In contrast, at 5 dpi all untreated animals displayed severe pulmonary impairments and we observed a dose-dependent effect of favipiravir (Figure 4b and 4d). When using a preemptive antiviral strategy, all animals treated with 37.5mg/day TID had marked damages and animals treated with 75mg/day TID displayed mild or moderate damages (Figure S5). When using a preventive antiviral strategy, all animals treated with 37.5mg/day TID had mild to marked damages and animals treated with 75mg/day TID displayed no or mild damages (Figure 4e-h). At 5 dpi, significant cumulative score reductions were observed with both doses of favipiravir regardless the therapeutic strategy used ($p=0.0286$, details in Table S8)."

Lines 205-225: The new figure 4 and its legend were added.

Line 233: "Figure 2a" was replaced by "Figure 1".

Lines 234-235: The sentence "Favipiravir was quantified in plasma (n=9) and lung tissue (n=3)." was replaced by "Favipiravir trough concentrations were quantified in plasma (n=9) and lung tissue (n=3).".

Line 236: "Figure S4" was replaced by "Figure S7".

Line 256: "Table S5" was replaced by "Table S9".

Lines 257-258: In the legend of the Table 3, the words ", at the end of the dosing interval (trough concentrations)" was added.

Line 263: "Figure 2" was replaced by "Figure 1".

Lines 269 and 272: "Table S6" was replaced by "Table S10".

Lines 272-275: The sentence "Overall, no majority mutations were detected (mutation frequency >50%), mutations were distributed throughout the whole genome and almost all of them exhibited a frequency lower than 10% (Figure 5a and 5b)." was replaced by "Overall, no majority mutations were detected (mutation frequency >50%) and almost all of the mutations exhibited a frequency lower than 10% mutations (Figure 5a). In addition, mutations were distributed throughout the whole genome (Figure 5b)."

Line 291: "Table S8" was replaced by "Table S13".

Lines 294-307: To increase the clarity of the original figure 5a, (i) we have added the sign « # » and the legend « Animal ID » and (ii) we have modified the y-axis numbering. Supplemental table related to the figure was updated (Table S10, S11, S12 and S13).

Line 311: The expression "followed by observation of significant weight losses" was removed.

Lines 312-313: The expression "weight loss" was replaced by "the lack of weight gain".

Line 313: The number "6" was replaced by "7".

Lines 314-316: The sentences "Histopathological changes are similar to those previously described. Notably, our results revealed that all animals with marked or severe pulmonary impairments displayed vascular lesions (endothelitis, vasculitis) as previously described in humans." were added.

Line 316: The word "Overall" was added.

Line 321: The reference "(Figure 1 and 2)" was added.

Lines 325-326: The words "and in hamsters treated with favipiravir" were added.

Lines 326-327: The sentence "Furthermore, the histopathological study revealed that favipiravir exhibit a protective role on the lungs, particularly noticed 5dpi." was added.

Line 330: The reference "(Figure 2)" was added.

Line 332: The word "losses" was removed.

Lines 334-348: The paragraph "In the present study, reduction of viral replication was correlated with the dose of favipiravir administrated and inversely correlated with the dose of virus inoculated. In a recent study, favipiravir administrated per os twice daily (loading dose of 600mg/kg/day followed by 300mg/kg/day) revealed a mild reduction of lung viral RNA yields using a similar hamster model with high doses of virus (2×10^6 TCID₅₀). These results are in accordance with ours at the lower dose of favipiravir (around 340mg/kg/day TID)." was modified in "In the present study, reduction of viral replication was correlated with the dose of favipiravir administrated and inversely correlated with the dose of virus inoculated. In a recent study, the efficacy of favipiravir intraperitoneally administrated per os twice daily (loading dose of 900 and 1,200mg/kg/day followed by 600 and 1,000mg/kg/day respectively) was assessed using a similar hamster model (6 to 10 weeks old) with high doses of virus (2×10^6 TCID₅₀). Treatment with the highest dose of favipiravir resulted in a moderate decrease of viral RNA yields in lung tissue and the lowest dose induced an even smaller inhibitory effect. However, significant infectious titers reduction were observed in a dose-dependent manner in lungs. Both doses were also associated with regression of pulmonary histopathological impairments. Overall, these results are in accordance with ours at the lower medium and the high dose of favipiravir (around 670 and 1390mg/kg/day TID). However, in this other study, no signs of toxicity were associated with favipiravir treatment regardless the dosing regimen. This discrepancy could be due to the difference between (i) the highest daily doses used (1,000mg/kg/day in regards to 1390mg/kg/day in our study), (ii) the dosing regimens (BID instead of TID in our study), and/or (iii) the age of the hamsters at day of infection (6 to 10 weeks old in comparison to 4 weeks old in our study)." .

Lines 349-350: The words "since it is recognized as a purine nucleotide by the viral RNA-dependent RNA polymerase." were added to the sentence.

Line 356: The words "and C→U" were removed.

Line 356: The reference "(Figure 5)" was added.

Lines 389-398: The paragraph "The medium dose of favipiravir used in this study (670mg/kg/day TID) is within the range of the estimated doses required to reduce by 90% (ED₉₀) the level of infectious titers in lungs (ranging between 570 and 780mg/kg/day). Animals treated with this dose displayed significant reduction of viral replication in lungs, limited drug-associated toxicity and clinical alleviation of the disease. Regarding the accumulation ratio after repeated doses and the good penetration of favipiravir in lungs, effective concentrations can be expected in lungs, throughout the course of treatment using this dose of 670mg/kg/day TID." was modified in "The medium dose of favipiravir used in this study (670mg/kg/day TID) is within the range of the estimated doses required to reduce by 90% (ED₉₀) the level of infectious titers in lungs (ranging between 31 and 42 mg/day corresponding to 570-780mg/kg/day) (Table 2) and displayed limited drug-associated toxicity (Figure 2b). Animals infected with 10^5 and 10^4 TCID₅₀ of virus, and treated following a preemptive strategy with this dose displayed significant reduction of infectious titers and histopathological damages in lungs and clinical alleviation of the disease (Figure 1, 2 and 4). Animal treated following a preventive strategy with this dose also displayed significant reduction of viral replication and histopathological impairments in lungs (Figure 3 and 4). Regarding the accumulation ratio after repeated doses and the good penetration of favipiravir in lungs, effective concentrations can be expected in lungs, throughout the course of treatment using this dose of 670mg/kg/day TID." .

Line 445: The concentrations of "78.5, 39.3, 19.6µg/mL" were replaced by "500, 250 and 125µM".

Lines 497-498: "Table S9" was replaced by "Table S14".

Lines 532-544: The paragraph “Histology. Animal handling, hamster infections and favipiravir administrations were performed as described above. Lungs were collected after intratracheal instillation of 4% (w/v) formaldehyde solution, fixed 72h at room temperature with a 4% (w/v) formaldehyde solution and embedded in paraffin. Tissue sections of 3.5µm, obtained following guidelines from the ‘global open RENI’ (The standard reference for nomenclature and diagnostic criteria in toxicologic pathology; <https://www.goreni.org/>), were stained with hematoxylin-eosin (H&E) and blindly analyzed by a certified veterinary pathologist. Microscopic examination was done using a Nikon Eclipse E400 microscope. Different anatomic compartments were examined (see Table S15): (1) for bronchial and alveolar walls, a score of 0 to 4 was assigned based on severity of inflammation; (2) regarding alveoli, a score of 0 to 2 was assigned based on presence and severity of hemorrhagic necrosis; (3) regarding vessel lesions (endothelitis/vasculitis), absence or presence was scored 0 or 1 respectively. A cumulative score was then calculated and assigned to a grade of severity (see Table S16).” was added.

Line 551: “Table S10” was replaced by “Table S17”.

Line 572: “Table S6” was replaced by “Table S10”.

Line 580: The words “Graphical representations and” were added.

Lines 584-585: The sentence “When relevant, two-sided statistical tests were always used.” was added.

Line 568: The sentence “Experimental timelines were created on biorender.com.” was added.

Lines 587-590: The paragraph “Data Availability. Raw sequence reads of the virus genome analysed in this study have been deposited in the BioProject data bank (PRJNA648821). Authors can confirm that all other relevant data are included in the paper and/or its supplementary information files.” was added.

Lines 593-594: The sentence “We thank Lionel Chasson (CIML; Marseille) for helping to generate low magnification pictures.” was added.

Lines 604-606: The initials of the new author were added: “C.L.”.

Lines 615-641: The names of the supplemental figures and tables were updated.

REVIEWERS' COMMENTS

Reviewer #1 (Remarks to the Author):

The authors were very responsive to the comments from the reviewers. They added additional data demonstrating that 37 and 75 mg/kg TID favipiravir reduces lung pathology. They included an extensive statistical analysis and clarified some of the language and figures.

Reviewer #2 (Remarks to the Author):

The authors made a serious effort to revise the manuscript.

I was also asked to comment on the revisions made in response to Reviewer #3, which I find overall satisfactory. Specific suggestions below.

Please further update the sentence "As of 6 November 2020, more than 48.8 million cases of COVID-19 have resulted in more than 1,235,000 deaths"

Lines 133-135: "High toxicity was observed with the dose of 75mg/day TID with significant weight loss noticed from the first day of treatment (Table S4)." was replaced by " High toxicity was observed with the dose of 75mg/day TID with, from the first day of treatment, normalized weights significantly lower than those of untreated animals (Table S5)."

The phrasing "high toxicity" is not standard language; this could for example be "Important toxicity" or "Significant signs of toxicity".

Line 142: The sentence "Each dose of virus was assessed in an independent experiment." was added

The word "dose" is more appropriate for drug dosing; for infection studies with viruses, I believe it is better to use 'virus inoculum' I suggest to make this change throughout the manuscript.

In the section on histology (somewhere between Lines 183-204) it is written “When using a preemptive antiviral strategy, all animals treated with 37.5mg/day TID had marked damages and animals treated with 75mg/day TID displayed mild or moderate damages (Figure S5). When using a preventive antiviral strategy, all animals treated with 37.5mg/day TID had mild to marked damages and animals treated with 75mg/day TID displayed no or mild damages (Figure 4e-h).”

I find this a somewhat strange phrasing. It should at least be mentioned that there are pathological changes in the lungs.

Lines 272-275: “Overall, no majority mutations were detected (mutation frequency >50%) and almost all of the mutations exhibited a frequency lower than 10% mutations (Figure 5a).

This is also a strange phrasing, I would suggest “... all of the mutations occurred at a frequency lower than...”

Lines 314-316: “Histopathological changes are similar to those previously described...”

Similar rather means identical. Better would be “comparable”

Lines 389-398: “... histopathological impairments in lungs...”

this should be rephrased; I think that functionalities can be impaired, but not histopathology scores.

**The reviewers,
*Nature Communications***

Marseille, January 11th 2021

Once again, we would like to address our gratitude to the reviewers for their noteworthy review of our work.

The following represents a reply to each of the comments and then refers to the changes made in the revised manuscript.

Reviewer #1 (Remarks to the Author):

The authors were very responsive to the comments from the reviewers. They added additional data demonstrating that 37 and 75 mg/kg TID favipiravir reduces lung pathology. They included an extensive statistical analysis and clarified some of the language and figures.

We thank the reviewer for the positive comments on our work.

Reviewer #2 (Remarks to the Author):

The authors made a serious effort to revise the manuscript.

I was also asked to comment on the revisions made in response to Reviewer #3, which I find overall satisfactory. Specific suggestions below.

We thank the reviewer for the positive comments on our work.

Please further update the sentence “As of 6 November 2020, more than 48.8 million cases of COVID-19 have resulted in more than 1,235,000 deaths”

The sentence was updated.

Lines 133-135: “High toxicity was observed with the dose of 75mg/day TID with significant weight loss noticed from the first day of treatment (Table S4).” was replaced by “ High toxicity was observed with the dose of 75mg/day TID with, from the first day of treatment, normalized

weights significantly lower than those of untreated animals (Table S5).” The phrasing “high toxicity” is not standard language; this could for example be “Important toxicity” or “Significant signs of toxicity”.

The word « high » was replaced by the word « important ».

Line 142: The sentence “Each dose of virus was assessed in an independent experiment.” was added. The word “dose” is more appropriate for drug dosing; for infection studies with viruses, I believe it is better to use ‘virus inoculum’ I suggest to make this change throughout the manuscript.

The expressions « virus inoculum » and « virus inocula » were added throughout the manuscript in order to replace the inappropriate use of the word « dose ».

In the section on histology (somewhere between Lines 183-204) it is written “When using a preemptive antiviral strategy, all animals treated with 37.5mg/day TID had marked damages and animals treated with 75mg/day TID displayed mild or moderate damages (Figure S5). When using a preventive antiviral strategy, all animals treated with 37.5mg/day TID had mild to marked damages and animals treated with 75mg/day TID displayed no or mild damages (Figure 4e-h).” I find this a somewhat strange phrasing. It should at least be mentioned that there are pathological changes in the lungs.

These sentences were replaced by the following sentences : « When using a preemptive antiviral strategy, all animals treated with 37.5mg/day TID had marked histopathological damages in lungs and animals treated with 75mg/day TID displayed mild or moderate histopathological damages (Figure S5). When using a preventive antiviral strategy, all animals treated with 37.5mg/day TID had mild to marked damages in lung and animals treated with 75mg/day TID displayed no or mild histopathological damages (Figure 4e-h). »

Lines 272-275: “Overall, no majority mutations were detected (mutation frequency >50%) and almost all of the mutations exhibited a frequency lower than 10% mutations (Figure 5a). This is also a strange phrasing, I would suggest “... all of the mutations occurred at a frequency lower than...”

The words « almost all of the mutations exhibited a frequency lower than 10% mutations » were replaced by the words « almost all of the mutations occurred at a frequency lower than 10% ».

Lines 314-316: “Histopathological changes are similar to those previously described...” Similar rather means identical. Better would be “comparable”.

The word « similar » was replaced by the word « comparable ».

Lines 389-398: “... histopathological impairments in lungs...” this should be rephrased; I think that functionalities can be impaired, but not histopathology scores.

The sentences « Animals infected with 10^5 and 10^4 TCID₅₀ of virus, and treated following a preemptive strategy with this dose displayed significant reduction of infectious titers and histopathological damages in lungs and clinical alleviation of the disease (Figure 1, 2 and 4). Animal treated following a preventive strategy with this dose also displayed significant reduction of viral replication and histopathological impairments in lungs (Figure 3 and 4). » were replaced by the sentences « Animals infected with 10^5 and 10^4 TCID₅₀ of virus, and treated following a preemptive strategy with this dose displayed significant reduction of infectious titers and histopathological scores in lungs and clinical

alleviation of the disease (Figure 1, 2 and 4). Animal treated following a preventive strategy with this dose also displayed significant reduction of viral replication and histopathological scores in lungs (Figure 3 and 4). ».